# Deciphering DED assembly mechanisms in FADD-procaspase-8-cFLIP complexes regulating apoptosis

Chao-Yu Yang[1], Chia-I Lien[2], Yi-Chun Tseng[1,3,6], Yi-Fan Tu[3,6], Arkadiusz W. Kulczyk[4], Yen-Chen Lu[3], Yin-Ting Wang[1], Tsung-Wei Su[1], Li-Chung Hsu [2,5,7] ✉, Yu-Chih Lo [3,7] ✉ & Su-Chang Lin [1,7] ✉

Fas-associated protein with death domain (FADD), procaspase-8, and cellular FLICE-inhibitory proteins (cFLIP) assemble through death-effector domains (DEDs), directing death receptor signaling towards cell survival or apoptosis. Understanding their three-dimensional regulatory mechanism has been limited by the absence of atomic coordinates for their ternary DED complex. By employing X-ray crystallography and cryogenic electron microscopy (cryo-EM), we present the atomic coordinates of human FADD-procaspase-8-cFLIP complexes, revealing structural insights into these critical interactions. These structures illustrate how FADD and cFLIP orchestrate the assembly of caspase-8-containing complexes and offer mechanistic explanations for their role in promoting or inhibiting apoptotic and necroptotic signaling. A helical procaspase-8-cFLIP hetero-double layer in the complex appears to promote limited caspase-8 activation for cell survival. Our structure-guided mutagenesis supports the role of the triple-FADD complex in caspase-8 activation and in regulating receptor-interacting protein kinase 1 (RIPK1). These results propose a unified mechanism for DED assembly and procaspase-8 activation in the regulation of apoptotic and necroptotic signaling across various cellular pathways involved in development, innate immunity, and disease.

The fundamental role of death receptor (DR) signaling pathways lies in making vital life-or-death decisions that are crucial for proper embryogenesis, tissue development, immune responses, and the maintenance of tissue homeostasis[1,2]. The formation of various signaling complexes through homotypic death domain (DD)-fold assembly is a pivotal process. Upon activation by death ligands (DL), receptors like Fas (CD95) and tumor necrosis factor-related apoptosis-inducing ligand receptors (TRAILR, including DR4 and DR5) recruit an adapter protein, Fas-associated death domain (FADD) via DD-DD interactions. In contrast, receptor-like tumor necrosis factor receptor 1 (TNFR1) recruits TNFR-associated death domain (TRADD). These DR initiate distinct signaling cascades by forming different signaling complexes[3,4]. Nevertheless, the precise mechanical underpinnings of this process remain unclear.

The CD95/TRAILR-FADD receptor complex assembles the death-inducing signaling complex (DISC), or complex I, by homotypic death-

[1]Genomics Research Center, Academia Sinica, Taipei 11529, Taiwan. [2]Institute of Molecular Medicine, College of Medicine, National Taiwan University, Taipei 10002, Taiwan. [3]Department of Biotechnology and Bioindustry Sciences, College of Bioscience and Biotechnology, National Cheng Kung University, Tainan 70101, Taiwan. [4]Institute for Quantitative Biomedicine, Rutgers University, Department of Biochemistry and Microbiology, Rutgers University, Piscataway, NJ 08854, USA. [5]Graduate Institute of Immunology, College of Medicine, National Taiwan University, Taipei 10002, Taiwan. [6]These authors contributed equally: Yi-Chun Tseng, Yi-Fan Tu. [7]These authors jointly supervised this work: Li-Chung Hsu, Yu-Chih Lo, Su-Chang Lin. ✉e-mail: lichunghsu@ntu.edu.tw; gracelo@ncku.edu.tw; tomlin@gate.sinica.edu.tw

effector domain (DED) assembly with procaspase-8 (Casp-8) and antiapoptotic cellular FLICE-inhibitory proteins (cFLIP)[5,6]. This complex is subsequently dismantled into cytosolic secondary complexes, known as complex II[7]. Both complex I and II can activate caspases, driving apoptosis[7,8]. In DL-sensitive cells, the levels of cFLIP are often insufficient to block DR-induced apoptosis[9–11].

The signaling complex formation within the TNFR1 pathway is more intricate. DD-mediated formation of TNFR1 complex I initiates receptor-interacting protein kinase 1 (RIPK1)-dependent NF-κB activation, leading to the expression of antiapoptotic proteins such as cFLIP for cell survival[12–21]. The subsequent internalization of TNFR1 complex I releases its signaling components, which bind to FADD, procaspase-8, and cFLIP to form cytosolic complex IIa[21], typically exerting an anti-apoptotic effect unless blocked by inhibiting the production of RIPK1-NF-κB-mediated antiapoptotic proteins[16,22–25], for instance, by cycloheximide (CHX)[26,27].

Within the FADD-Casp-8-cFLIP subcomplex, cFLIP plays a pivotal role in determining cell fate[6,28,29]. Two cFLIP isoforms, cFLIP_L and cFLIP_S, differentially regulate Casp-8 activation[30]. Both isoforms can form a heterodimer with Casp-8, inhibiting the formation of fully active Casp-8 heterotetramers and thus apoptosis[6,31,32]. Notably, the FADD-Casp-8-cFLIP_L complex can cleave RIPK1, further suppressing necroptosis, RIPK1-mediated apoptosis, and inflammatory responses[33–37].

Evidently, both FADD-Casp-8 and FADD-Casp-8-cFLIP complexes are summoned in regulating DR-mediated apoptosis and RIPK1-mediated necroptosis. However, the precise structural mechanisms governing FADD-Casp-8-cFLIP DED assembly remain largely unclear. The cryo-EM structure of Casp-8 tandem DED (Casp-8[tDED]) filament (PDB: 5L08)[38] reveals interactions limited to Casp-8[tDED]-Casp-8[tDED] interactions. Moreover, some EM envelops of the multiprotein DED complex were constructed at 12–15 angstrom resolutions (EMDB: EMD-11939 and EMD-11941)[39], which proved insufficient for building critical atomic coordinates[40,41]. The absence of atomic coordinates or a PDB file that unequivocally illustrates multiprotein DED assembly further compounds the mystery.

DED, a member of the DD-fold superfamily, plays a key role in regulating signaling within innate immunity, inflammation, and programmed cell death[42]. In mammals, each single DD-fold typically features six surfaces for homotypic interactions, including type Ia, type Ib, type IIa, type IIb, type IIIa, and type IIIb surfaces. Presumably, upstream single DDs transiently self-assemble to create a composite-binding site (CBS) comprising type Ia, type IIa, and type IIIa surfaces for recruiting downstream DDs to form cryptic helical complexes like the Myddosome and PIDDosome[43,44]. However, downstream Casp-8 and cFLIP both have a tandem DED (tDED), in contrast to a single DED of FADD. The specific FADD[DED]-FADD[DED], FADD[DED]-Casp-8[tDED], FADD[DED]-cFLIP[tDED], Casp-8[tDED]-cFLIP[tDED], and cFLIP[tDED]-cFLIP[tDED] interactions in DED assemblies remain unclear.

Despite several models proposed for the multiprotein DED assembly[38,39,45–49], the mystery persists regarding how FADD, Casp-8, and cFLIP utilize DED for assembly. Atomic coordinates, which are invaluable for revealing and elucidating the intricacies of DED assembly, are notably absent.

In this work, we present the crystal and cryo-EM structures of ternary FADD-Casp-8-cFLIP DED complexes, unveiling FADD-FADD, FADD-Casp-8, FADD-cFLIP, Casp-8-cFLIP, and cFLIP-cFLIP DED assemblies in a complex. The atomic coordinates shed light on the mechanical mechanism by which DED assembly influences cell fate. The atomic coordinates depict how FADD[DED] and Casp-8[tDED] could assemble a helical 1:5 and 3:3 intermediate complexes, and how these FADD-Casp-8 intermediate complexes are targeted and capped by four cFLIP[tDED] molecules. The resultant helical Casp-8[tDED]-cFLIP[tDED] hetero-double layer appears to locally activate Casp-8. Mutagenesis results suggest that the triple-FADD complex regulates TNF-mediated apoptotic signaling and RIPK1 activation and cleavage.

## Results

### Enhancing complex reconstitution by Casp-8[tDED] mutations
In order to explore the interactions between FADD, Casp-8, and cFLIP that regulate signaling, we generated various combinations to reconstitute the DED core complexes for our structural studies. However, when Casp-8[tDED] was overexpressed, it predominantly formed filaments, regardless of the presence of cFLIP or full-length FADD (FADD[FuL]), and precipitated in *E. coli*. This precipitation hindered the reconstitution of multiprotein complexes (Fig. 1a).

Mutations in Casp-8[tDED] were employed to increase the population of monomers[38,45,47,50]. For instance, a double mutation, F122A/L123A, targeting the conserved hydrophobic phenylalanine/leucine motif (the FL motif, Supplementary Fig. 1a) on the type Ib surface of Casp-8 DED2, prevented the formation of the death effector filament (DEF) or self-filamentation when overexpressed in HeLa cells[47]. Therefore, we applied a similar strategy to enrich the population of soluble tDED complexes. Notably, the overexpression of the Casp-8[tDED] F122A mutant (Casp-8[tDED_F122A]) still led to the formation of apoptotic DEFs[51] and efficient recruitment to the DR5 DISC but impaired TRAIL-induced apoptosis[45]. We observed that co-expression of Casp-8[tDED_F122A] and cFLIP[tDED] H7G mutant (CF[H7G]), both of which lack the protease domain, led to a mixture of Casp-8[tDED_F122A] DEF filaments and oligomeric CF[H7G]-Casp-8[tDED_F122A] tDED complexes (Fig. 1b). This suggests that cFLIP is capable of inhibiting Casp-8 filamentation but is outcompeted by Casp-8 self-filamentation during overexpression studies.

To further reduce Casp-8 self-filamentation during the reconstitution of DED core complexes, we introduced F122G/L123G double mutations, often referred to as FGLG. Importantly, the Casp-8[tDED_F122GL123G] mutant (C8[FGLG]) is predominantly monomeric in gel filtration[38] and exhibits no self-filamentation (Fig. 1c). However, it is capable of forming short filaments in the presence of FADD[FuL] (Fig. 1c). This observation implies that FADD can still interact with C8[FGLG] to induce C8[FGLG] filamentation.

Collectively, the results suggest that Casp-8 with mutations in the FL motif or F122 only affect Casp-8[tDED]-Casp-8[tDED] assembly in Casp-8[tDED] filament extension but not FADD[DED]-Casp-8[tDED] and Casp-8[tDED]-cFLIP[tDED] assemblies in the FADD-Casp-8-cFLIP complex. Therefore, these mutations can facilitate the reconstitution of multiprotein DED complexes to unveil these unknown assemblies. As demonstrated by the atomic coordinates we present below, the interaction between FADD and Casp-8 utilizes the type Ib surface of Casp-8 DED2 rather than its FL motif. In addition, the interaction between cFLIP and Casp-8 does not involve the FL-motif containing, type Ib surface of Casp-8.

### The structure of the single-FADD-Casp-8-cFLIP DED complex
We successfully reconstituted the ternary complex FADD[FuL_H9G]-Casp-8[tDED_F122G/L123G]-cFLIP[tDED_H7G] complex, henceforth referred to as the FA[FuL_H9G]-C8[FGLG]-CF[H7G] complex (Supplementary Fig. 2a). Subsequently, we obtained the protein crystals and determined the crystal structure at 3.1 Å resolution using the MR-SAD method (Supplementary Table 1 and Supplementary Figs. 2b and 3), in which FADD DD is not visible. Notably, the crystal structure reveals a 1:5:4 stoichiometry for the FA[FuL_H9G]-C8[FGLG]-CF[H7G] complex (Fig. 2a, c, Supplementary Movie 1). This complex represents a single-FADD ternary DED complex, which fits the corresponding SAXS envelope (Supplementary Figs. 2c and 4 and Supplementary Table 2).

In the crystal structure, FADD is sub-stoichiometric to Casp-8, consistent with previous observations in CD95 and TRAILR DISCs[46,47]. Our recombinant ternary complex is in agreement with the TRAILR DISC from BJAB cells, which contains a single FADD subunit and a 1:9 ratio of FADD to tDED-containing proteins[47]. If a TRAILR DISC comprises three pairs of TRAILR and TRAIL, a single FADD, and nine full-length Casp-8 molecules, it would have a molecular weight of ~790 kDa. This aligns with the characterization of a high molecular weight (HMW) DISC, which is >700 kDa, in tumor cells[47]. Therefore,

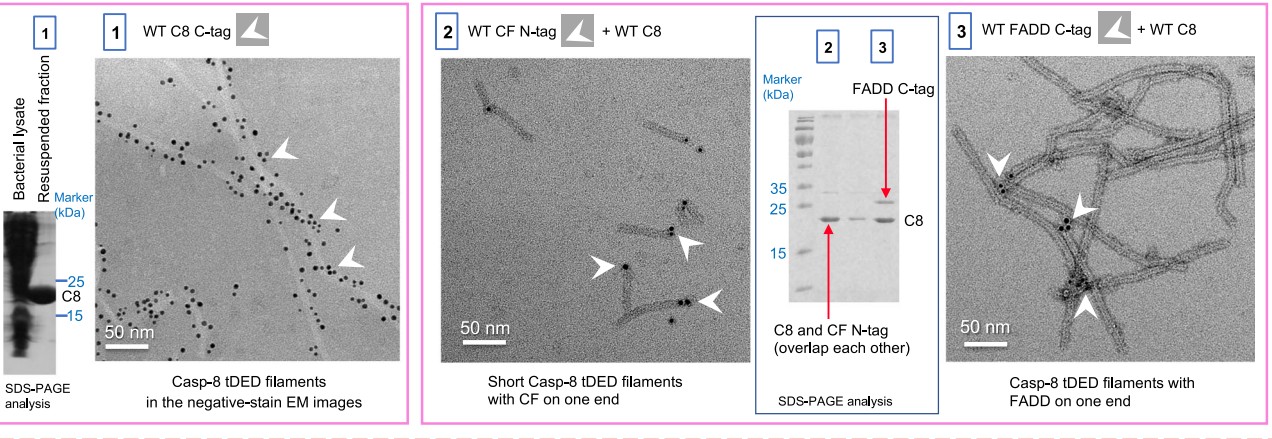

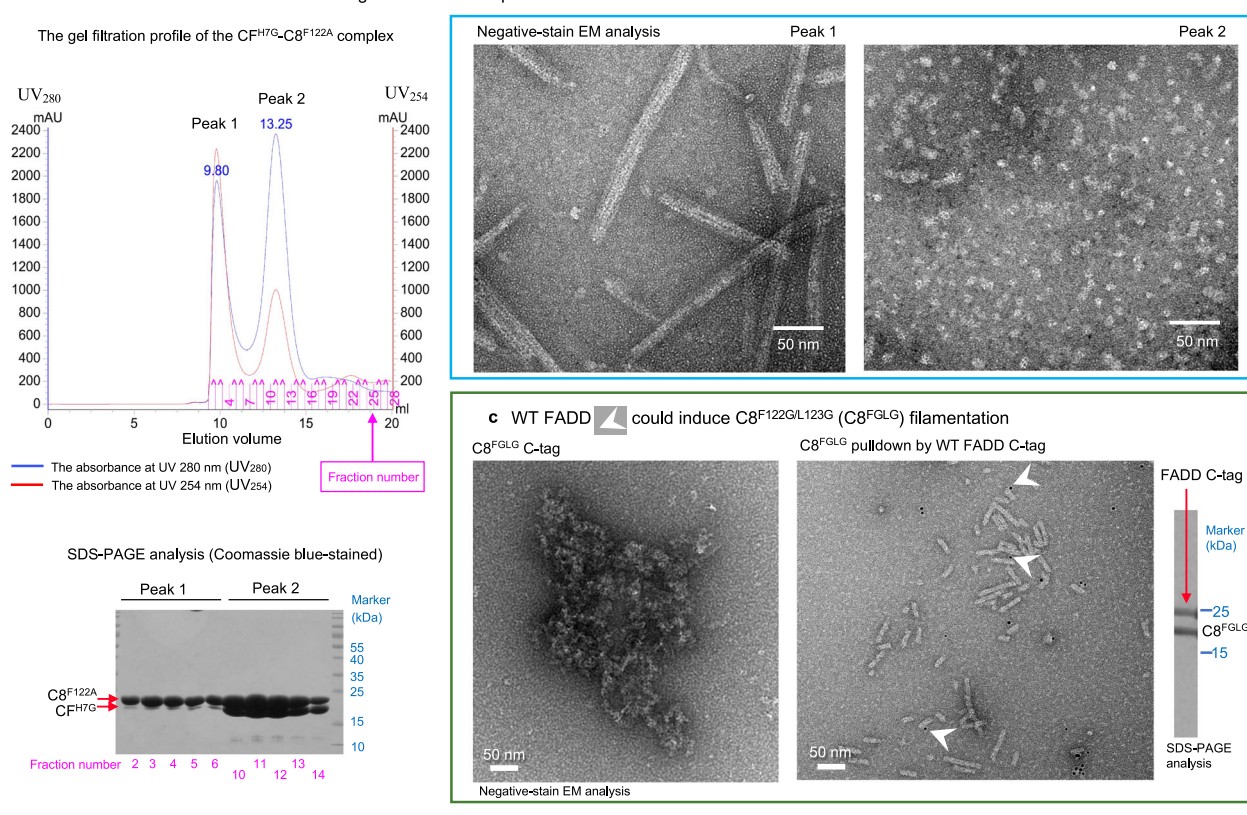

**Fig. 1 | Casp-8^tDED mutations impair Casp-8 self-filamentation but facilitate complex reconstitution. a** The wild-type (WT) Casp-8^tDED (C8) predominantly forms filaments. Negative-stain electron microscopy (EM) and Coomassie blue-stained SDS-PAGE analysis of precipitates formed by the wild-type protein or complexes and resuspended in water. White arrowheads indicate some His-tagged proteins detected by Nanogold. Sample 1: WT Casp-8^tDED with a C-terminal His-tag (WT C8 C-tag). Sample 2: WT Casp-8^tDED plus WT cFLIP^tDED with an N-terminal His-tag (WT CF N-tag). Sample 3: WT Casp-8^tDED plus WT FADD with a C-tag. **b** Casp-8^tDED_F122A mutant aids in the reconstitution of soluble DED complexes. SEC profile of the binary CF^H7G-Casp-8^tDED_F122A complex and negative-stain EM and SDS-PAGE analyses

of the peak fractions. AU absorbance units. **c** Monomeric Casp-8^tDED_F122G/L123G mutant (C8^FGLG) forms short filaments in the presence of WT FADD. Negative-stain EM images show that C8^FGLG purified by SEC cannot form filaments, whereas it forms short filaments in the presence of WT FADD, although FADD mostly dissociated. White arrowheads indicate C-terminal His-tagged (C-tag) FADD detected by Nanogold. The SDS-PAGE analysis of the EM sample of the binary complex is also shown. All micrograph and biochemical data were repeated independently twice with similar results. Scale bar = 50 nm. Source data are provided as a Source Data file.

our single-FADD ternary DED complex closely resembles the DED subcomplex within the TRAILR DISC.

## The structure of the triple-FADD-Casp-8-cFLIP DED complex

To delve deeper into the assembly of the FADD-Casp-8-cFLIP complex, we conducted a study using wild-type (WT) full-length FADD, WT cFLIP^tDED, and a full-length Casp-8 mutant with FGLG mutations, catalytic inactivity and processing deficiencies (F122G/L123G/C360A/

D374A/D384A). This allowed us to reconstitute a ternary complex known as the FA^FuL-C8^FuL_FGLG_CADA-CF complex with an intact Casp-8 protease domain (Supplementary Fig. 2d). We then employed single-particle cryo-electron microscopy (cryo-EM) to examine this complex in detail (Supplementary Fig. 5, 6).

Upon our initial examination without crosslinking, we obtained a cryo-EM structure of the FA^FuL-C8^FuL_FGLG_CADA-CF complex at 7.5 Å resolution, referred to as Complex A (Supplementary Fig. 5, 6 and

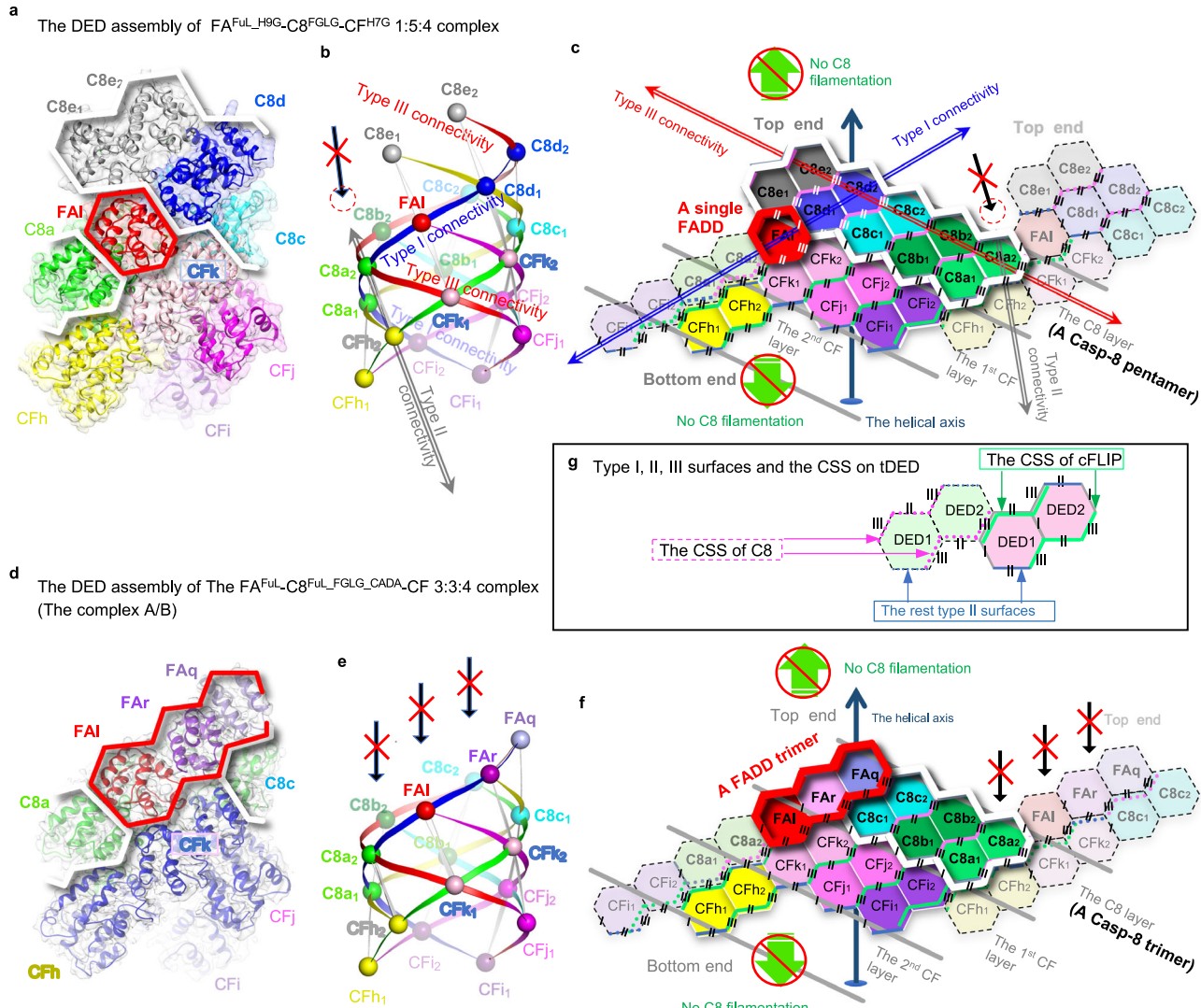

**Fig. 2 | FADD, Casp-8, and cFLIP assemble a single- and a triple-FADD complexes. a–c** The ribbon, ball-and-ribbon, and hexagon diagrams in **a–c**, respectively, depict the crystal structure of the single-FADD ternary complex. In **a** and **c**, the FADD and Casp-8 portions are highlighted by red and white thick lines, respectively. In **b**, the diagram shows the type I, II, and III connectivity by ribbons connecting every two adjacent DEDs, represented by balls (Supplementary Fig. 1c, "Methods" section). In **b** and **c**, a red dashed circle indicates the adjoining space of Casp-8 molecule C8e, while a long black arrow with a red X indicates the space was not filled by any DED. Each hexagon in the 2D representation represents a DED, with six sides representing the type I, II, and III surfaces. Each protein or tDED molecule is labeled with its chain ID and colored uniquely. CFk1, for example, stands for the DED1 of cFLIP (CF) with the chain ID k. The molecule with the same chain ID retains consistent coloring across all three diagrams to facilitate comparison and identification, except that Casp-8 and cFLIP are colored in green and blue, respectively, in **d** and Fig. 7b. The dashed hexagons, which adjoin the tDED with the solid lines, represent an adjacent tDED from a different layer. The angled and angled/dashed lines highlight the CSS of tDEDs with solid edges and dashed edges, respectively. Notably, cFLIP and Casp-8 CSS are colored in green and pink, respectively. Top and bottom ends, see Supplementary Fig. 13. **d–f** Similar to **a–c**, except that the diagrams **d–f** depict the cryo-EM structure of the triple-FADD ternary complex. Long black arrows with a red X indicate that FADD physically blocks Casp-8 filamentation on the top end. **g** Shows a type III interaction between cFLIP^tDED and Casp-8^tDED, colored in pink and green, respectively.

Supplementary Table 2). Unexpectedly, this structure revealed a 3:3:4 ratio of FADD-Casp-8-cFLIP DED core complexes, forming what we refer to as a triple-FADD ternary DED complex. However, both Casp-8 protease domain and FADD DD were not visible, suggesting that these two domains didn't form a stable complex.

To validate this observation, we produced a GraFix-stabilized version of the FA^FuL-C8^FuL_FGLG_CADA-CF complex and solved the cryo-EM structure at 3.7 Å resolution, named Complex B (Supplementary Figs. 5–9 and Supplementary Table 3). Complex B confirmed the 3:3:4 model observed in Complex A (Fig. 2d, f, Supplementary Fig. 2e, and Supplementary Movie 2), with invisible Casp-8 protease domain and FADD DD.

Both Complex A and Complex B exhibit a balanced ~1:1:1 ratio of FADD, Casp-8, and cFLIP within the complex. This ratio aligns with previous observations made in CD95 DISC from cFLIP_S-overexpressing BJAB and HaCaT cells[48] and SKW6.4 cells[46] and also in DR5 DISC from A549 and H460 cells[45]. These findings support the notion that our triple-FADD ternary complex closely resembles the DED subcomplexes found in these DISCs.

## Casp-8^tDED utilizes type III-II-III CSS to self-assemble

It's known that FADD recruits Casp-8 through a hierarchical binding process, ultimately forming a ternary complex with cFLIP that can inhibit CD95- and TRAIL-mediated apoptotic signaling[48]. The crystal structure of the ternary 1:5:4 FA^FuL_H9G-C8^FGLG-CF^H7G complex and the cryo-EM structure of the ternary 3:3:4 FA^FuL-C8^FuL_FGLG_CADA-CF complex provide insights into two different FADD-Casp-8 intermediate

complexes in DR-mediated signaling. The former suggests the formation of a single-FADD-Casp-8 intermediate complex, with one FADD and five Casp-8 molecules (Fig. 3a, c and Supplementary Movie 3). The latter proposes a triple-FADD-Casp-8 intermediate complex, involving three FADD and three Casp-8 molecules (Fig. 3b, c).

In both the single-FADD and triple-FADD complexes, Casp-8$^{tDED}$ consistently utilizes a composite-self-assembling site (CSS) composed of a type II and two type III surfaces (Fig. 2c, f, g), referred to as the type III-II-III CSS (Supplementary Fig. 1b), to form a transient left-handed pentamer (Supplementary Movie 3) and trimer, respectively (Fig. 3c). This emphasizes the importance of the type III-II-III CSS in Casp-8$^{tDED}$

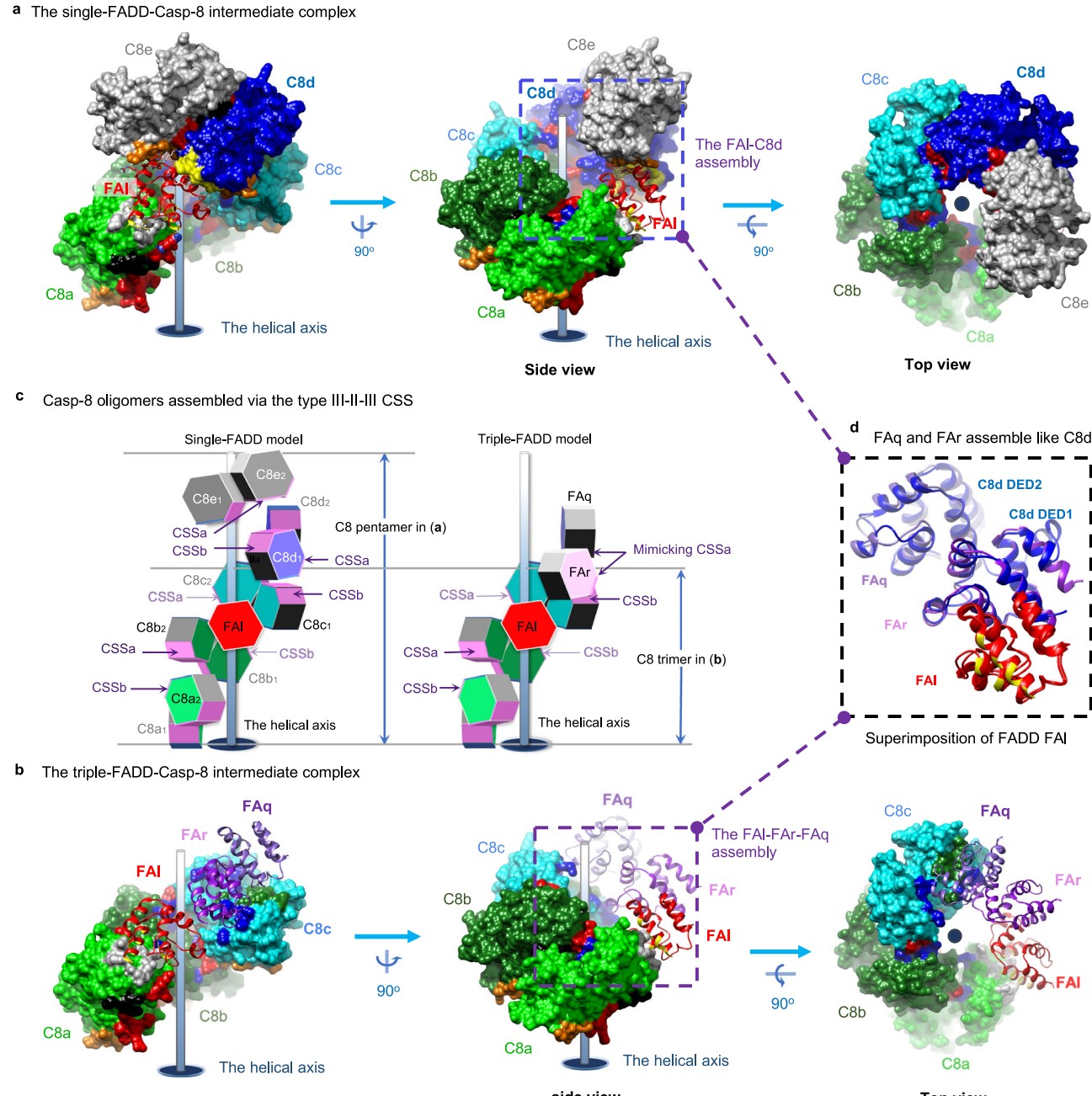

**Fig. 3 | Casp-8$^{tDED}$ utilizes the type III-II-III CSS to self-assemble in both intermediate complexes. a** Shows the single-FADD-Casp-8 intermediate complex derived from the single-FADD ternary complex structure by removing cFLIP. FADD$^{DED}$, as ribbons, and Casp-8$^{tDED}$, as molecular surfaces, are labeled and colored as in Fig. 2a. Type Ia, IIb, IIIa, and IIIb surfaces of Casp-8$^{tDED}$ are colored in yellow, green, red, and blue, respectively, similar to those in Supplementary Fig. 1. However, for clarity, the type Ib surface of Casp-8$^{tDED}$ molecule C8a is colored in gray, while type IIa surfaces of Casp-8 DED1 and DED2 are colored in orange and black, respectively. **b** Similar to **a**, except that here shows the triple-FADD-Casp-8 intermediate complex derived from the triple-FADD ternary complex structure. **c** Side-by-side comparison of the assembly of the single- and triple-FADD-Casp-8

intermediate complexes. The comparison reveals that Casp-8$^{tDED}$ assembles consistently via CSS, which is highlighted in magenta. Additionally, the lower parts of two intermediate complexes consistently contain FADD$^{DED}$ molecule FAI and Casp-8$^{tDED}$ molecules C8a-c. However, in the upper part of the triple-FADD complex, FADD$^{DED}$ molecules FAq and FAr could together assemble a surface mimicking the CSSa of the Casp-8$^{tDED}$ molecule C8d in the single-FADD complex, also explained in **d**. **d** Structural comparison between the single-FADD-single-Casp-8 assembly, i.e., the FAI-C8d assembly, in **a** and the triple-FADD assembly in **b** by superimposing FADD$^{DED}$ molecule FAI. The comparison reveals that FADD$^{DED}$ molecules FAq and FAr could assemble like a Casp-8$^{tDED}$ molecule C8d and hence form a surface that mimics CSSa in **c**.

self-assembly and filament formation, as supported by our results showing that C8[tDED_FGLG] can still form short filaments in the presence of FADD (Fig. 1c).

Furthermore, the type III-II-III CSS-mediated Casp-8[tDED] trimer in our triple-FADD structure (Fig. 2f and Supplementary Fig. 10a) aligns with the Casp-8[tDED] trimer found in the actual asymmetric unit of the Casp-8[tDED] filament (PDB: 5L08)[38], as portrayed in Supplementary Fig. 11a. Therefore, the minimum repeating unit for assembling Casp-8[tDED] filaments is a type III-II-III CSS-mediated Casp-8[tDED] trimer.

## FADD[DED] and Casp-8[tDED] assemblies in intermediate complex

Our triple-FADD complex structure unveils FADD[DED]-FADD[DED] and FADD[DED]-Casp-8[tDED] assemblies (Fig. 4a), by which three WT FADD[DED] molecules FAl, FAq, and FAr form a right-handed, type I interface-mediated trimer (Fig. 4a, b) to interact with the left-handed Casp-8[tDED] trimer assembled by Casp-8[tDED] molecules C8a, C8b, and C8c (Fig. 4a, c). Notably, FADD[DED] interacts with the type Ib surface of C8[tDED_FGLG] DED2 (Fig. 4b), although we introduced FGLG mutations on the type Ib surface, suggesting that the FL motif is dispensable for FADD[DED]-Casp-8[tDED] assembly.

In addition, two FADD[DED] molecules FAq and FAr (Fig. 4d) assemble like the Casp-8[tDED] molecule C8d in the single-FADD complex (Fig. 4f). Together they provide a type II and two type III surfaces, mimicking the type III-II-III CSS, for interaction with Casp-8[tDED] molecule C8c (Fig. 4c, d vs. 4e, f). Therefore, to form a triple-FADD complex, the FADD trimer could sequentially recruit three Casp-8[tDED] molecules C8c, C8b, and then C8a, via the type III-II-III CSS, as portrayed in Stage 3 of Supplementary Fig. 12a. Alternatively, the FADD trimer could bind simultaneously C8a and C8c of a transient, type III-II-III CSS-mediated Casp-8[tDED] trimer, which contains C8a, C8b, and C8c, and stabilize the trimer (Stage 3 of Supplementary Fig. 12a). The FADD[DED]-FADD[DED] and FADD[DED]-Casp-8[tDED] interactions unveiled by the atomic coordinates of the triple-FADD complex (Fig. 4a) allow us to propose how FADD[DED] and Casp-8[tDED] assemble a triple-FADD intermediate complex in DR-mediated signaling (Supplementary Fig. 12a).

## FADD-Casp-8 assembly in the single-FADD intermediate complex

The single-FADD complex structure unveils additional FADD[DED]-Casp-8[tDED] assembly, in which Casp-8[tDED] assembles a type III-II-III CSS-mediated pentamer, containing Casp-8[tDED] molecules C8a, C8b, C8c, C8d, and C8e (Fig. 3a). Although the only FADD[DED] molecule FAl uses distinct surfaces to interact with three different Casp-8[tDED] molecules C8a, C8d, and C8e, these interfaces are too small, which are ~330 to ~500 Å², to form 1:1 FADD-Casp-8 complexes for intermediate complex formation (Supplementary Fig. 14a, b). Instead, the III-II-III CSS contributes larger interface areas, which are ~780 to ~820 Å² corresponding to the buried surface area (the BSA) of ~1560 to ~1640 Å², providing Casp-8[tDED] the ability to self-assemble transient oligomers. These oligomers then form a composite-binding site (CBS) with a larger interface area of 835 - 1,280 Å² to interact effectively with FADD (Supplementary Fig. 14c), promoting the formation of a FADD-Casp-8 intermediate complex (Supplementary Fig. 12b and Supplementary Movie 3, 4).

In summary, both triple-FADD and single-FADD structures argue that Casp-8[tDED] utilizes the type III-II-III CSS to self-assemble functional oligomers (Supplementary Fig. 12a, b). This may play a significant role, especially considering the abundance of Casp-8 in various cell types[9,52,53] with levels higher than FADD and cFLIP[11,46,54]. It is plausible that high levels of Casp-8 could promote the formation of transient Casp-8 oligomers, contributing to the creation of a FADD-Casp-8 intermediate complex.

It's important to note that endogenous Casp-8 typically does not form filaments automatically in cells; instead, it requires FADD upon DR stimulation. It was believed that a conformational change in FADD was triggered to expose its DED to seed the formation of Casp-8 DED

complex[39,52,55]. Our structures argue that the role of exposed FADD DED in seeding may involve stabilizing transient Casp-8 oligomers assembled via the type III-II-III CSS of tDED to complete a FADD-Casp-8 intermediate complex. This intermediate complex can subsequently recruit cFLIP to form an anti-apoptotic ternary complex (Supplementary Movie 5) or Casp-8 to create apoptotic filaments in DR-mediated signaling (Supplementary Fig. 12). This insight explains why C8[FGLG] requires FADD to form an intermediate complex for initiating C8[FGLG] filamentation (Fig. 1c).

## cFLIP targets the CBS on FADD-Casp-8 intermediate complex

cFLIP plays a crucial role in inhibiting CD95- and TRAIL-mediated apoptotic signaling by binding to the FADD-Casp-8 complex[48]. Since our structures unveil FADD[DED]-cFLIP[tDED], Casp-8[tDED]-cFLIP[tDED], and cFLIP[tDED]-cFLIP[tDED] assemblies (Fig. 2), our structural analysis could provide detailed insights into how cFLIP targets this complex in preventing the formation of Casp-8 filaments (Fig. 5).

In both intermediate complexes, the first cFLIP[tDED] that binds the FADD-Casp-8 intermediate complexes should be cFLIP[tDED] molecule CFk because its interface area, which is >1,650 Å² corresponding to a BSA of >3,300 Å² with ~38 hydrogen bonds/salt bridges, is much larger than those of cFLIP[tDED] molecules CFi and CFj, which are ~730 Å² (Fig. 2c, f). Therefore, our structures demonstrate that the FADD-Casp-8 intermediate complexes assemble a composite-binding site (CBS) to recruit CFk (Fig. 5a).

In the 1:5:4 FA[FuL_H9G]-C8[FGLG]-CF[H7G] complex structure, three Casp-8[tDED] molecules C8c, C8d, and C8a provide, respectively, a type Ia, a type IIa, and a type IIIa surfaces, while FADD[DED] molecule FAl provides a type IIa and a type IIIa surfaces to complete the CBS to recruit cFLIP[tDED_H7G] molecule CFk (Fig. 5a, c, d). A similar CBS, which recruits WT cFLIP[tDED] molecule CFk (CFk-binding CBS), was also found in the 3:3:4 FA[FuL]-C8[FuL_FGLG_CADA]-CF complex structure, although Casp-8[tDED] C8d is replaced by two WT FADD[DED] molecules FAr and FAq (Fig. 5c vs 5e).

Once an upstream cFLIP[tDED] molecule, for example, CFk, binds the CBS, it, in turn, provides surfaces to assemble the next CBS for recruiting the next cFLIP[tDED] molecule CFj (Fig. 5b, Supplementary Movie 6). The CBS for recruiting the second cFLIP[tDED] molecule CFj consists of a type III-II-III CSS from cFLIP[tDED] molecule CFk and a type Ia and a type IIa surfaces respectively from two Casp-8[tDED] molecules C8b and C8c, with an interface area of ~1,400 Å² in total (Fig. 5b). The third cFLIP molecule CFi is recruited in a similar way along the type III connectivity (Fig. 5b), via a CBS with an interface area of ~1200 Å². Therefore, the recruitment of cFLIP molecules, either cFLIP_L or cFLIP_S, would depend on a hierarchical process. In our structures, cFLIP[tDED] also forms a type III-II-III CSS-mediated trimer.

The CBS for recruiting the fourth cFLIP[tDED] molecule CFh, with an interface area of ~1300 Å², looks different from aforementioned CBS and has different Casp-8[tDED]-cFLIP[tDED] and cFLIP[tDED]-cFLIP[tDED] interactions. Two upstream cFLIP[tDED] molecules CFk and CFi provide a type Ia and a type IIIa surfaces, respectively, while Casp-8[tDED] molecule C8a provides two type IIa and a type IIIa surfaces (Fig. 5b). Notably, despite variations in CBS assembly, all cFLIP[tDED]-recruiting CBSs are composed of five surfaces, including a type Ia, two type IIa, and two type IIIa surfaces.

Compared to the cryo-EM structure of the Casp-8 filament[38], the region corresponding to where cFLIP[tDED] molecule CFk occupies in our structures actually binds a Casp-8[tDED] (Supplementary Fig. 11a), although via a smaller interface area of ~1,437 Å² with fewer hydrogen bonds/salt bridges of ~27 (Fig. 5f). Therefore, the CBS on the FADD-Casp-8 intermediate complex probably would prefer cFLIP[tDED] to Casp-8[tDED]. This preference aligns with previous studies that have shown cFLIP's hierarchical targeting of the FADD-Casp-8 complex to regulate DR- or RIPK1-induced cell death signaling[29,30,48]. Importantly, cFLIP's binding effectively caps the bottom end of the Casp-8[tDED] layer in both complexes, physically preventing further extension (Fig. 2c, f, Supplementary Fig. 13, Supplementary Movie 1, 2).

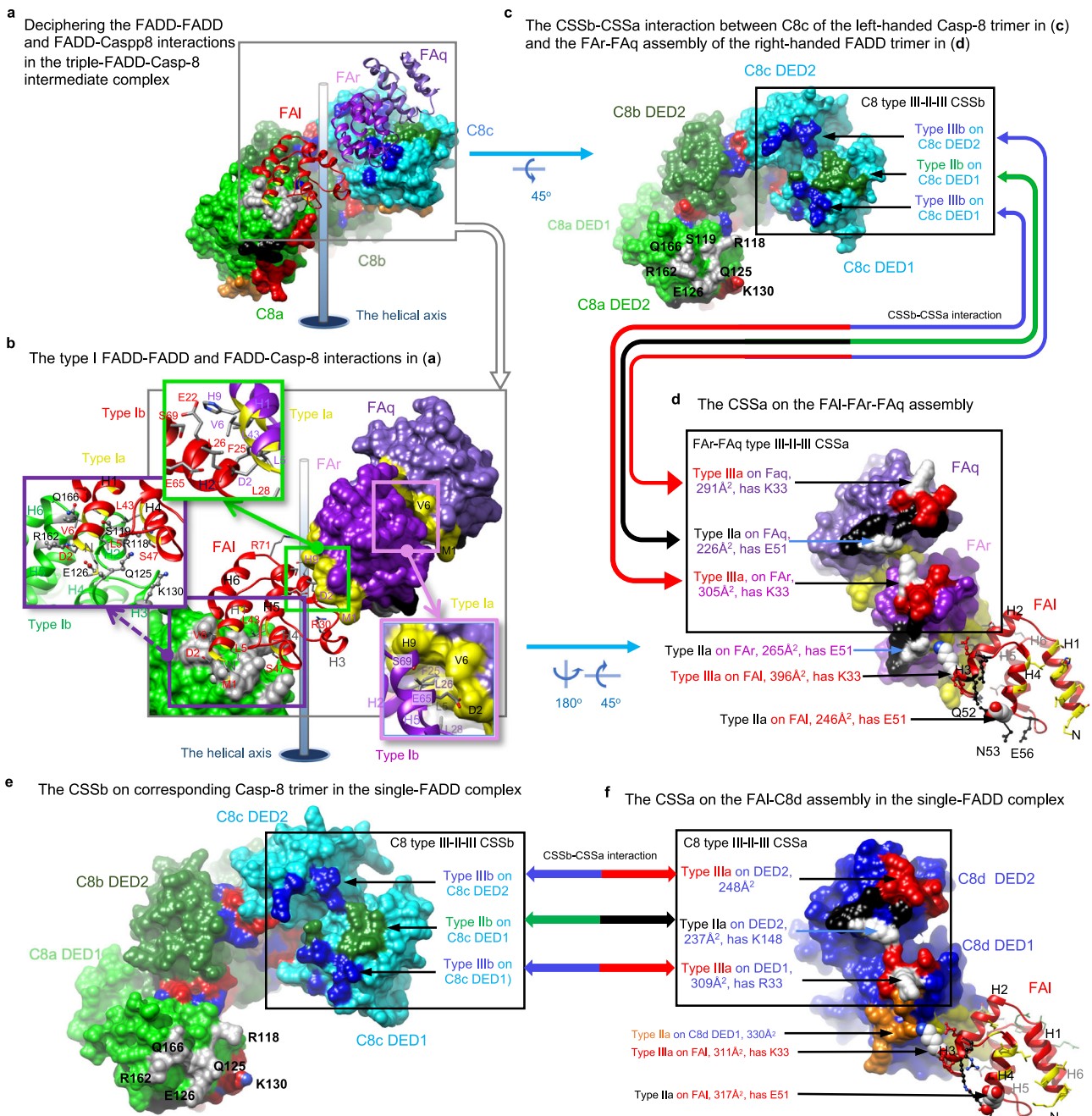

**Fig. 4 | FADD$^{DED}$-FADD$^{DED}$ and FADD$^{DED}$-Casp-8$^{tDED}$ assemblies in the intermediate complex. a** Shows that FADD$^{DED}$ molecules FAl, FAr, and FAq assemble a right-handed trimer that binds a left-handed Casp-8$^{tDED}$ trimer in the triple-FADD-Casp-8 intermediate complex. The molecules are labeled and colored as in Fig. 3a, b. FAq and FAr refer to FADD$^{DED}$ (FA) with the chain ID q and r, respectively. **b** Shows the interface residues on type I FADD$^{DED}$-FADD$^{DED}$ and FADD$^{DED}$-Casp-8$^{tDED}$ interfaces. The green and pink boxes show the residues on the type I interfaces in the triple-FADD, i.e., the FAl-FAr-FAq assembly, while the purple box shows the interface residues between Casp-8$^{tDED}$ molecule C8a and FADD$^{DED}$ molecule FAl. The type Ia surfaces and corresponding ribbons of FADD$^{DED}$ are colored in yellow. The interface residues of FADD$^{DED}$ molecules FAl, FAr, and FAq are labeled in red, purple, and black, respectively. while those on Casp-8$^{tDED}$ molecule C8a are labeled in black.

**c, d** Black boxes and double-headed arrows highlight the CSSb-CSSa interaction between the Casp-8$^{tDED}$ trimer (**c**) and FADD$^{DED}$ trimer (**d**) of the triple-FADD complex. In **d**, type IIa and IIIa surfaces are colored in black and red, respectively, in which functionally important FADD residues K33 and E51 are highlighted in white. The size of each interface on FADD$^{DED}$ is shown in parentheses. Notably, in **c**, the type Ib residues of Casp-8$^{tDED}$ molecule C8a are labeled in black. **e-f**, Black boxes and double-headed arrows highlight the corresponding CSSb-CSSa interaction in the single-FADD complex, between the corresponding Casp-8$^{tDED}$ trimer (**e**) and single-FADD-single-Casp-8 assembly (**f**) with similar orientations to the those in **c** and **d**, respectively. Notably, in **f**, Casp-8 surfaces are colored as in Fig. 3a, except that functionally important Casp-8 residues R33 and K148[38] have white surfaces.

## Casp-8-cFLIP$_L$ hetero-double layer locally activates Casp-8

Intriguingly, both the WT cFLIP$^{tDED}$ and the cFLIP$^{tDED}$ mutant yield a ternary complex structure with four cFLIP$^{tDED}$ molecules. In addition, the 3:3:4 and 1:5:4 FADD-Casp-8-cFLIP complex structures consistently exhibit an almost 1:1 ratio of cFLIP to Casp-8 (Fig. 2c, f). This ratio echoes previous research, which emphasizes the significance of heterodimerization between the protease domains of Casp-8 and cFLIP for local Casp-8 activation[48,56–58]. Importantly, our structures unveil the addition of a cFLIP$^{tDED}$ layer atop the Casp-8$^{tDED}$ layer within the FADD-Casp-8 intermediate complex, thus creating a Casp-8-cFLIP hetero-

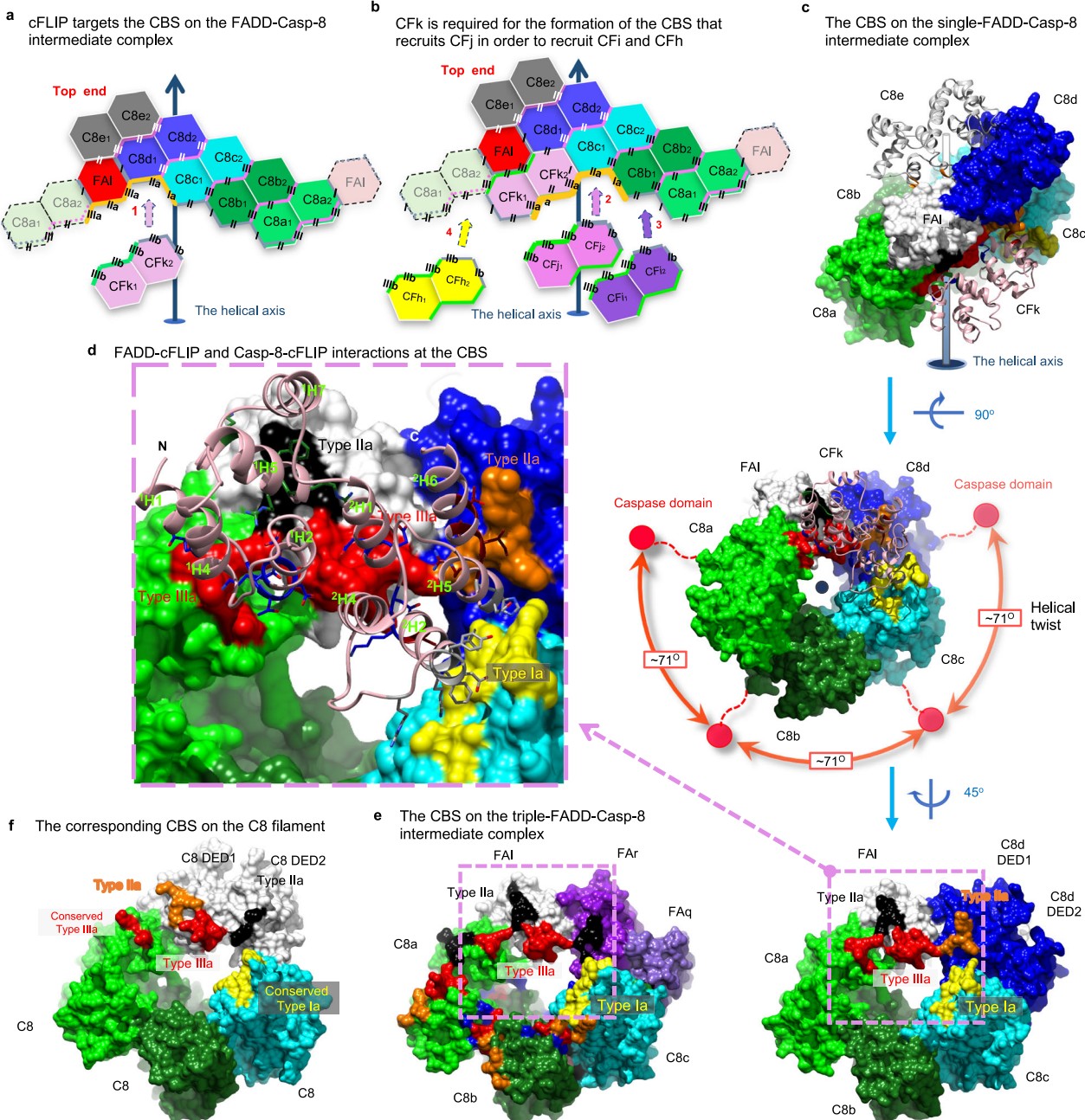

**Fig. 5 | cFLIP targets the CBS of the FADD-Casp-8 intermediate complex and further caps it. a** Thick orange lines highlight the CBS of the single-FADD intermediate complex shown in 2D representation. The CBS is targeted by cFLIP$^{tDED}$ molecule CFk, as the first step of cFLIP recruitment. **b** Thick orange lines highlight the CBS generated after the recruitment of cFLIP$^{tDED}$ molecule CFk to the single-FADD complex. The CBS could recruit another cFLIP$^{tDED}$ molecule CFj. This process repeats until the complex is capped by cFLIP and reaches a maximum capacity of four cFLIP molecules. **c** Shows FADD$^{DED}$-cFLIP$^{tDED}$ and Casp-8$^{tDED}$-cFLIP$^{tDED}$ assemblies in the single-FADD complex. Surface representations of the single-FADD intermediate complex viewed from different angles show that the CBS contains a type Ia surface (yellow) from Casp-8$^{tDED}$ molecule C8c DED1, a type IIa surface (orange) from Casp-8$^{tDED}$ molecule C8d DED1, a type IIa (black) and a type IIIa (red) surfaces from FADD$^{DED}$ molecule FAI, and a type IIIa surface (red) from Casp-8$^{tDED}$ molecule C8a DED2. Red balls in the middle panel indicate the possible locations of the protease domains. **d** The pink box highlights FADD$^{DED}$-cFLIP$^{tDED}$ and Casp-8$^{tDED}$-cFLIP$^{tDED}$ interactions in **c**. cFLIP$^{tDED}$ molecule CFk is shown as pink ribbons, while its interface residues are shown as sticks. In **c–e**, FADD$^{DED}$ and Casp-8$^{tDED}$ are labeled and colored as in Fig. 2a, except that FADD$^{DED}$ molecule FAI is colored in white for clarity and for the comparison with the white Casp-8$^{tDED}$ molecule in **f**. **e** The pink box highlights the corresponding CBS surfaces in the triple-FADD intermediate complex. Notably, two type IIa surfaces (black) are from FADD$^{DED}$ molecules FAI and FAr. **f**, The corresponding CBS surfaces in the C8 filament (5L08 [https://doi.org/10.2210/pdb5L08/pdb])[38]. White surfaces highlight the C8 molecule that provides two type IIa (black and orange) and a type IIIa (red) surfaces.

double layer (Fig. 2a–f). This configuration beautifully elucidates how the tDED complex formation facilitates the heterodimerization between cFLIP and Casp-8.

Furthermore, because cFLIP$_S$ lacks a protease domain, the assembly of a Casp-8$^{tDED}$-cFLIP$^{tDED}$ hetero-double layer unequivocally allows only the FADD-Casp-8-cFLIP$_L$ complex, but not FADD-Casp-8-cFLIP$_S$, to foster heterodimerization between the protease domains of cFLIP and Casp-8, resulting in local Casp-8 activation for processes like cleaving RIPK1 to inhibit necroptosis[33]. Consequently, our structures also clarify structurally why the recruitment of cFLIP$_L$, but not cFLIP$_S$, to the FADD-Casp-8 intermediate complex (Fig. 2) leads to partial/local Casp-8 activation.

Notably, in contrast to Casp-8 in DEF that could fully activate Casp-8 to generate apoptotic Casp-8 heterotetramers[6,31,32], Casp-8 in the FADD-Casp-8-cFLIP$_S$ complex mentioned earlier is inactive. Hence, a single layer of Casp-8, consisting of either a trimer or pentamer, in our FADD-Casp-8 intermediate complexes appears unable to trigger Casp-8 autoactivation. This is likely due to the tDED complex's helical twist of approximately 71 degrees along the type III connectivity, which may hinder the protease domain of Casp-8, shown as prominent red spheres in Fig. 5c, from homodimerization. Therefore, our structures further argue that both cFLIP$_L$ and cFLIP$_S$ binding to the FADD-Casp-8 intermediate complex would prevent the addition of a second Casp-8 layer and, consequently, the full activation of Casp-8.

In addition, we can postulate that, during DR activation, a shortage of cFLIP would allow the formation of a Casp-8 double layer, setting the stage for the dimerization of Casp-8's protease domains. As mentioned earlier, the type III-II-III CSS-mediated Casp-8$^{tDED}$ trimer represents the minimum repeating unit (Supplementary Fig. 11a) in the assembly of Casp-8$^{tDED}$ filaments. A cFLIP shortage-mediated generation of a Casp-8 double layer, comprised of two non-apoptotic Casp-8 trimers stacked along the type I connectivity, emerges as a pivotal step in achieving full Casp-8 activation in FADD-mediated or Casp-8 filamentation-mediated apoptotic pathways.

## Two unidirectional FADD-mediated Casp-8 filamentation models

Significantly, our EM results reveal that FADD and cFLIP attach to the Casp-8$^{tDED}$ filaments only at one end (Fig. 1a). However, in our structures, they occupy opposite ends (Fig. 2d). These observations suggest that FADD-induced Casp-8 filamentation is unidirectional, leaving the other end open for cFLIP to physically obstruct it, as discussed earlier. In fact, the triple-FADD structure demonstrates that FADD can physically impede Casp-8 filamentation at the upper end (Fig. 2e, f). In the single-FADD structure, the halt of Casp-8 filamentation at the upper end results from the inability of the CSS of Casp-8$^{tDED}$ molecule C8e to bind and stabilize another Casp-8 tDED when there's no FADD in the adjoining space (Fig. 2b, c). Consequently, our structures contribute to our understanding of the mechanism behind FADD-mediated unidirectional Casp-8 filamentation.

To assemble a Casp-8 double layer, we hypothesized that the CFk-binding CBS on the FADD-Casp-8 intermediate complex should interact with Casp-8 tDED (Supplementary Fig. 10a vs. 10b and 10c vs. 10d). Among the five surfaces that assemble the CBS, a type IIIa and a type Ia surfaces are offered, respectively, by two conserved Casp-8$^{tDED}$ molecules C8a and C8c (Supplementary Fig. 10a or 10c), which correspond to Casp-8$^{tDED}$ molecules C8g and C8b in Casp-8$^{tDED}$ filament structure (PDB: 5L08)[38] that contains no FADD (Supplementary Fig. 11a), with interface areas of approximate 200 and 500 Å$^2$ (Fig. 5f vs. 5c or 5e).

In addition, a type IIa and a IIIa surfaces are offered by FADD$^{DED}$ molecule FAl to bind cFLIP$^{tDED}$ molecule CFk, similar to how FADD$^{DED}$ molecule FAq binds Casp-8$^{tDED}$ molecule C8c in the 3:3:4 FADD-Casp-8-cFLIP complex structure (Supplementary Fig. 10a). The former FADD$^{DED}$ molecule contributes an interface area of around 630 Å$^2$, while the latter contributes 500 Å$^2$. Therefore, at least four surfaces of the CBS are capable of binding Casp-8$^{tDED}$. These four surfaces contribute 1,200 Å$^2$ of interface area, sufficient for binding a Casp-8$^{tDED}$, although no structure shows that the remaining type IIa surface of FADD$^{DED}$ molecule FAr or Casp-8$^{tDED}$ molecule C8d DED1, depicted as black dotted circles in Supplementary Fig. 10b and 10d, respectively, can bind Casp-8 DED2.

Thus, we propose that during DR activation, in the event of cFLIP shortage, the available Casp-8 in cells would be like Casp-8$^{tDED}$ molecule C8$_1$ that targets the CFk-binding CBS on the remaining FADD-Casp-8 intermediate complexes (Supplementary Fig. 10b or 10d), which is like cFLIP$^{tDED}$ would normally do (Supplementary Fig. 10a or 10c). The recruitment of Casp-8$^{tDED}$ molecule C8$_1$ would create a CBS

identical to the corresponding one in Casp-8 filaments. Consequently, the newly created CBS would recruit the second and third Casp-8, which are Casp-8$^{tDED}$ molecules C8$_2$ and C8$_3$ in Supplementary Fig. 10b, respectively, to complete the second type III-II-III CSS-mediated Casp-8 trimer and hence a Casp-8 double layer.

Importantly, the resultant Casp-8 double layer would also contain a newly created CBS that would initiate the recruitment of more Casp-8, such as Casp-8$^{tDED}$ molecule C8$_4$ and beyond, to form the third Casp-8 layer (Supplementary Movie 7). This process generates numerous Casp-8 helical double layers in filament extension for apoptotic Casp-8 activation. By substituting cFLIP$^{tDED}$ in our structures with Casp-8$^{tDED}$, we could establish both single-FADD and triple-FADD models (Supplementary Fig. 10b, d) for FADD-mediated unidirectional Casp-8 filamentation.

## Triple-FADD model regulates apoptotic signaling in cells

As it's uncertain whether TNFR, CD95, and TRAILR employ similar signaling mechanisms and complexes to activate Casp-8, we aimed to investigate the involvement of our models in hierarchical FADD-mediated Casp-8 activation. To do this, we created FADD-knockout (FADD-KO) iMEF cells using the CRISPR-Cas9 system (Supplementary Fig. 15a) and reconstituted FADD with different mutations on the five FADD-tDED interfaces identified in the single-FADD structure. These mutations included type Ia (Q40A), type IIa (E51R), type IIIa (R34A, K33E, E37A), type Ib (E22R, F25R, F25Y), and type IIb (D74A) mutations, indicated by the red, cyan, black, green, and orange boxes in Fig. 6a. We aimed to determine whether these mutations would impair apoptotic signaling.

However, only few FADD-deficient clones survived, and some of them expressed minimal RIPK3 or MLKL, e.g., the clones sg4.4, sg4.5, and sg4.10 in Supplementary Fig. 15a. This could be partially explained by previous observations that the embryonic lethality of FADD-deficient mice could be rescued by ablating RIPK3 to block necroptotic signaling[59]. Hence, the surviving FADD-KO iMEF clones may have a defect somewhere in the FADD-independent necroptotic signaling pathway. It's worth noting that surviving FADD-deficient MEF cells are known to be resistant to TNF-induced necroptosis[60,61]. We selected FADD-KO clone sg4.B9 for further study since it expressed both RIPK3 and MLKL (Supplementary Fig. 15a, lane 2), making it suitable for testing FADD-dependent apoptotic signaling and FADD-dependent RIPK3 activation.

In FADD-KO clone sg4.B9, RIPK1 and MLKL could be phosphorylated upon TNF/CHX treatment combined with the caspase inhibitor zVAD-fmk (zVAD) (TCZ treatment), as shown in Fig. 6c (lanes 1 vs. 2 and 13 vs. 14). However, subsequent FADD-independent necroptotic cell death induced by TNF was impaired (Supplementary Fig. 15e, column Vec). This finding is consistent with previous observations[60,61]. Importantly, although FADD-deficient Jurkat cells underwent RIPK3-dependent necroptosis upon TNF treatment[62,63], it is possible that different cells behave differently[64]. In clone sg4.B9, TNF-induced RIPK1-mediated IKK, p38, and JNK activation was observed similarly to iMEF WT cells (Supplementary Fig. 15g, h). However, TNF/CHX-induced Casp-8 activation was absent in clone sg4.B9, as expected (Supplementary Fig. 15b, lane 5).

Next, we reconstituted WT FADD and FADD mutants in the FADD-deficient iMEF cells to assess their impact on TNF/CHX-induced apoptosis. Unlike WT FADD, FADD F25R, K33E, E37K, E51R, and R34A mutants hardly restored TNF/CHX-induced Casp-8 activation, as indicated by less production of cleaved Casp-3 (c-Casp-3) (Fig. 6b). Among those mutants, FADD F25R, K33E, and E51R mutants hardly restored TNF/CHX-induced apoptosis (Supplementary Fig. 15c) and Casp-3 activation (Supplementary Fig. 15d). Casp-3 activation was confirmed by zVAD inhibition (Supplementary Fig. 15f). These results suggest that the type IIa (E51), type IIIa (K33), and type Ib (F25) surfaces of FADD are essential for Casp-8 activation and, consequently, apoptosis.

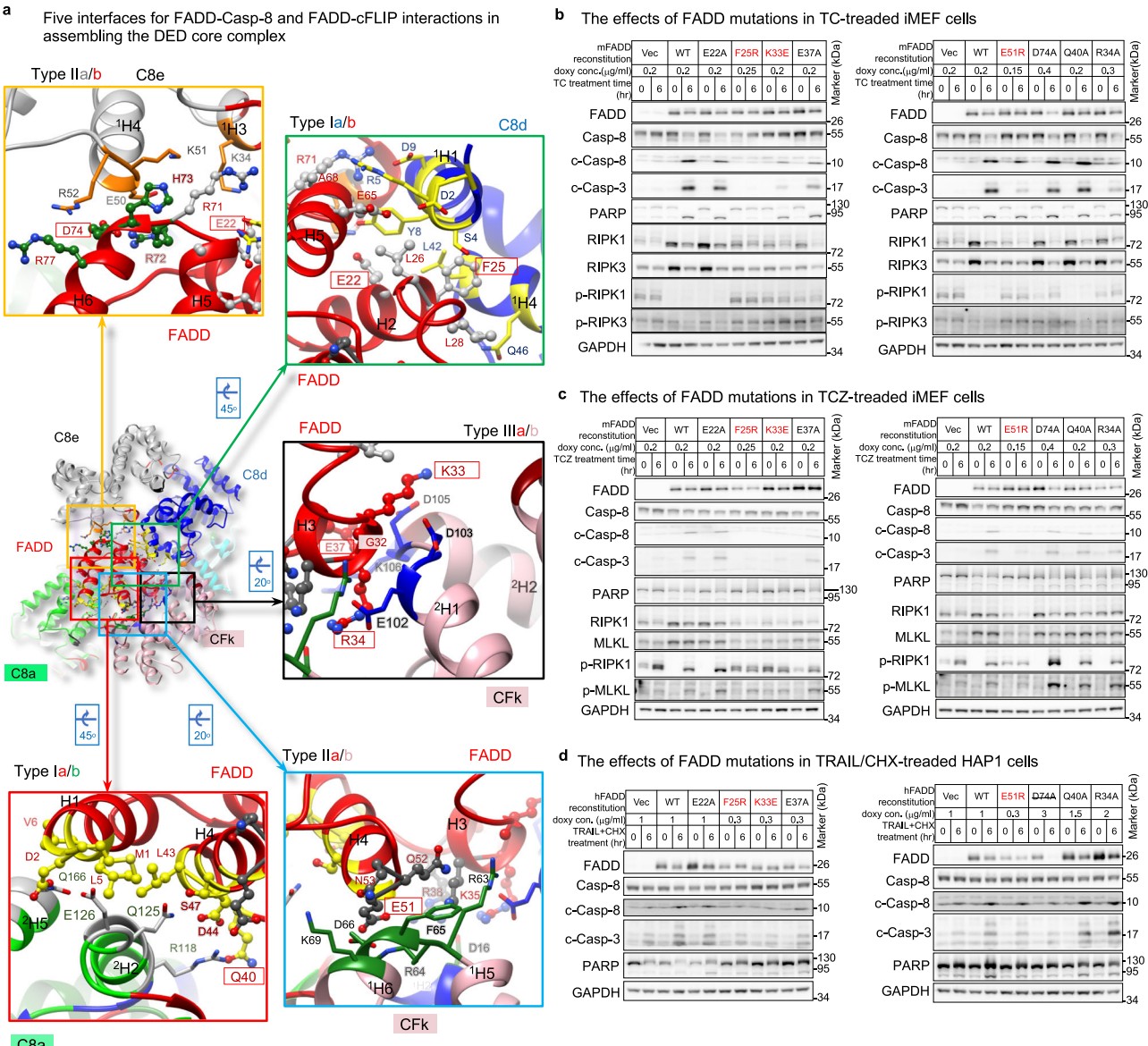

**Fig. 6 | Roles of the triple- and single-FADD models in hierarchical apoptotic signaling. a** Show five interfaces between FADD[DED] and tDED in the single-FADD structure. The interface residues of FADD are label in red, while FADD residues for mutagenesis are in red boxes. The side chains of the interface residues of FADD are shown as ball-and-sticks, while those of Casp-8 and cFLIP are shown as sticks. The interface residues of Casp-8 are colored as in Fig. 3a, while those of cFLIP are colored as in Supplementary Fig. 1a, respectively. The helices are numbered in Supplementary Fig. 2e. **b** Cell-based mutagenesis assays. FADD-reconstituted iMEF cells were treated with TNF/CHX (TC) to trigger apoptotic signaling. T TNFα, C CHX, Vec empty vecto, doxy doxycycline. F25R, K33E, and E51R mutants are highlighted in red because they are inactive in Supplementary Fig. 15c–f. **c** Same as **b**, but treated with TNF/CHX/zVAD (TCZ) to inhibit caspase activity and promote necroptotic signaling. Z zVAD. **d** Same as **b**, but FADD-reconstituted HAP1 cells were treated with TRAIL/CHX. All cell-based experiments were repeated twice with similar results. Uncropped blots are provided as a Source Data file.

Furthermore, because FADD D74 is located on type IIb surface (Fig. 6a, orange box) and is associated only with the single-FADD model, assessing the impact of the FADD D74A mutant on signaling would determine the necessity of FADD D74 and the single-FADD model. The results indicate that the FADD D74A mutant behaves like the WT protein (Fig. 6b, lane 20 vs. 16, Supplementary Fig. 15a, b), suggesting that FADD D74 and the single-FADD model are dispensable for TNF/CHX-induced TNFR1-mediated apoptotic pathways. Consequently, it is inferred that only the triple-FADD model is implicated in these pathways.

We also determined that FADD residues F25, K33, and E51 play crucial roles in TRAIL-induced Casp-8 activation in HAP1 cells (Fig. 6d). However, it remains inconclusive which model or whether FADD D74 was utilized, as the FADD D74A mutant in HAP1 cells was not stable

after death induction for an unknown reason (Fig. 6d, lane 20). It is noteworthy that, although a somewhat similar trend is observed in TC- and TCZ-treated iMEF cells (Fig. 6b, c, lane 20), FADD D74A mutation did not impact Casp-3 activation, apoptosis, and necroptosis (Supplementary Fig. 15c–e).

We further assessed the ability of the FADD proteins to activate Casp-8 in a HeLa cell lysate-based system, which has little cFLIP (Supplementary Fig. 15i, lane 1). Notably, aside from FADD F25Y, F25R, K33E, E37A, and E51R mutants, D74A mutant also failed to activate Casp-8 (Supplementary Fig. 15i, lane 11). This indicates that Casp-8 activation induced by added FADD requires FADD D74, type IIb surface, and potentially the single-FADD model. Noteworthily, FADD D74 is involved in CD95-mediated apoptosis in type II Jurkat and MCF7-Fas cells, or in forming the CD95-FADD-MC159 ternary complex[65–67].

## Triple-FADD complex regulates RIPK1 activation and cleavage

FADD-Casp-8-containing complexes not only regulate DR-induced apoptotic signaling but also have an impact on RIPK1-mediated necroptosis. In certain contexts, Casp-8 can cleave RIPK1, contributing to DL-induced apoptosis[68]. However, the inhibition of Casp-8 by zVAD preserves RIPK1 in these complexes, promoting RIPK1-dependent necroptosis triggered by ligands like FasL and TNF/CHX, particularly in primary T cells and Jurkat cells. Importantly, FasL/zVAD-induced necroptosis is FADD-dependent[62]. Furthermore, FADD appears to have a role in regulating TNF-induced necroptosis, as reintroducing full-length FADD into FADD-deficient Jurkat cells suppresses TNF-induced necrosis[69]. FADD is also known to negatively regulate TCR-mediated necroptosis[64]. To investigate whether the triple-FADD complex influences RIPK1 in TNF-induced necroptotic signaling, we evaluated the effects of FADD mutations in iMEF cells.

Upon TNF/CHX treatment, we observed a significant reduction in RIPK1 levels in iMEF cells expressing WT FADD or FADD E22A, R34A, Q40A, or D74A mutants (Fig. 6b, lanes 4, 6, 24, 22, and 20, respectively). This suggests that TNF-induced triple-FADD complexes activate Casp-8 to cleave RIPK1. In contrast, co-treatment with zVAD facilitated the induction of RIPK1 and MLKL phosphorylation (Fig. 6c, lanes 4, 6, 24, 22, and 20), promoting necroptotic signaling (Supplementary Fig. 15e). Collectively, these results indicate that TNF-induced apoptotic signaling involves the triple-FADD-Casp-8 complex, which leads to both Casp-8 activation and RIPK1 cleavage. Casp-8 inhibition, however, licenses the same complex to promote RIPK1 activation, thus facilitating necroptotic signaling. Notably, FADD-independent necroptosis does not occur in this setting, as discussed earlier.

## Discussion

We have successfully obtained the atomic coordinates (Fig. 2a–f) of ternary FADD-Casp-8-cFLIP DED complexes from 3.1 Å electron density maps and 3.7 Å cryo-EM envelops sufficient to atomic model building[40,41]. The atomic coordinates provide intricate insights into the specific DED assemblies involving FADD$^{DED}$-FADD$^{DED}$, FADD$^{DED}$-Casp-8$^{tDED}$, FADD$^{DED}$-cFLIP$^{tDED}$, Casp-8$^{tDED}$-cFLIP$^{tDED}$, and cFLIP$^{tDED}$-cFLIP$^{tDED}$ within the ternary complexes. Type III-II-III CSS is important for tDED to self-assemble, promoting the formation of the FADD-Casp-8 intermediate complex to generate the CBS for cell fate determination (Figs. 3–5 and Supplementary Fig. 10). These structural insights are critical for understanding cFLIP's hierarchical control over the FADD-Casp-8 complex in 3D, thereby regulating DR- or RIPK1-induced cell death signaling[29,30,48], as detailed in Fig. 7a–c.

The structural breakthrough was achieved by Casp-8$^{tDED}$ FGLG mutations and overexpressing cFLIP$^{tDED}$, effectively limiting Casp-8$^{tDED}$ self-filamentation. Consequently, the ternary DED complexes with a minimum DED assembly was purified homogeneously, leading to improved resolution and atomic model building[40,41]. The resultant minimum DED complex likely mirrors its WT Casp-8 tDED counterpart at certain stage, supported by previous observations where the complex of Casp-8 FGLG mutant is like WT Casp-8 that could cleave RIPK1, possibly in complex with cFLIP$_L$[70]. Notably, Casp-8 tDED FGLG mutations shield other intrinsic DED-DED interactions in the ternary complexes, allowing the identification of CFk as the sole cFLIP$^{tDED}$ molecule that interacts simultaneously with FADD$^{DED}$ and Casp-8$^{tDED}$ (Fig. 5a, b). Replacing cFLIP$^{tDED}$ molecule CFk (Supplementary Fig. 10a) with a Casp-8$^{tDED}$ (Supplementary Fig. 10b) would eliminate the sole FADD$^{DED}$-cFLIP$^{tDED}$ interaction in the complex, potentially initiating apoptosis.

In our atomic coordinates, DED2 of Casp-8$^{tDED}$ molecule C8a binds to FADD$^{DED}$ molecule FAl, followed by its DED1 binding to DED2 of cFLIP$^{tDED}$ molecule CFi along type I interactions (Fig. 2c, f). Therefore, earlier models, which propose Casp-8 DED1 binds to FADD$^{DED}$, followed by DED2 binding to the DED1 of the next Casp-8$^{tDED}$ molecule[39], may feature reversed orientations of DED2 to DED1 (Supplementary Fig. 11g

and Supplementary Movie 8). Alternatively, our structures offer a different model in which cFLIP$^{tDED}$ targets the CBS of the FADD-Casp-8 intermediate complex, with four cFLIP$^{tDED}$ molecules effectively obstructing the bottom-end extension of a Casp-8$^{tDED}$ trimer (Fig. 5a, b and Supplementary Fig. 13). Notably, since our structures and Casp-8$^{tDED}$ filament structure (PDB: 5L08)[38] comprise only type III-II-III CSS-mediated trimer (Supplementary Fig. 10a, c and 11a), it suggests that Casp-8$^{tDED}$ may dimerize along type III connectivity to form non-apoptotic type III-II-III CSS dimers (Supplementary Fig. 11b), called offset dimer previously[39], and trimers in the FADD-Casp-8 intermediate complex. Random dimerization along type II and III connectivity could yield three additional Casp-8$^{tDED}$ trimers (Supplementary Fig. 11c–e). However, these three trimers would generate apoptotic type II-III-II CSS dimers, also called aligned dimers previously[39], under non-apoptotic condition. Therefore, it is likely that an apoptotic type II-III-II CSS Casp-8$^{tDED}$ dimer, such as the Casp-8$^{tDED}$ C8h-C8a dimer (Supplementary Fig. 11a, b, f) and the Casp-8$^{tDED}$ dimer between N$^{th}$ and (N + 2)$^{th}$ asymmetric units (Fig. 7c), has to appear after the fourth Casp-8$^{tDED}$ molecule targets the FADD-Casp-8 intermediate complex (Fig. 7b) for a tighter control of apoptosis (Supplementary Movie 7). Future investigations, or a high-resolution map, are necessary to determine whether this model requires further modification.

The observed discrepancy in mutagenesis results indicates that the role of FADD residue F25 differs from that of FADD residue K33, E37, E51, or R34 (Fig. 6b and Supplementary Fig. 15c, d). A single mutation at F25 alters ~20% of the type IIa surface of FADD, a significantly greater change than the ~7% alteration of the CBS surface caused by a single mutation at K33, E37, E51, or R34. The F25R mutation likely disrupts the formation of the triple FADD-Casp-8 intermediate complex. Consequently, both Casp-7 and Casp-3 remain inactive, abolishing subsequent PARP cleavage (Fig. 6b). Conversely, mutations at K33, E37, E51, or R34 lead to partial Casp-3 activation (Fig. 6b), indicating that the FADD-Casp-8 intermediate complex assembles normally, but the ~7% change in the CBS surface results in partial Casp-8 activation. However, PARP cleavage could be attributed to the partial colocalization of Casp-7 and TNF-R1 receptosome[71], rendering Casp-7 activation and subsequent PARP cleavage insensitive to the ~7% change in the CBS. Additionally, Casp-7 exhibits better efficiency than Casp-3 in cleaving PARP[72,73]. Therefore, adding FADD to HeLa cell lysates could bypass TNF-R1 receptosome, making PARP cleavage sensitive to Casp-3 activation and CBS alterations (Supplementary Fig. 15i). These findings strongly link the formation of both the triple-FADD intermediate complex and CBS to critical processes such as caspase activation, apoptosis, and necroptosis.

Notably, adding WT FADD to HeLa cell lysates not only triggered Casp-8 activation but also stimulated RIPK1 activation (Supplementary Fig. 15i, lane 2). This unique activation of RIPK1 was disrupted by Casp-8 activation-defective FADD mutants (Supplementary Fig. 15i), reminiscent of the impact of Casp-8 knockdown on FADD-RIPK1 interaction[74]. However, the D74A mutant did not abolish RIPK1 activation (Supplementary Fig. 15i, lane 11), implying that the addition of FADD induces two functionally different complexes. One is involved in FADD D74-dependent Casp-8 activation, while the other, potentially the triple-FADD-Casp-8 complex, is involved in FADD D74-independent RIPK1 activation, which necessitates further investigation.

It's important to acknowledge that our structures do have some limitations. Because the FGLG mutations and overexpressing cFLIP$^{tDED}$ suppress Casp-8$^{tDED}$ self-filamentation in reducing sample heterogeneity and trapping a homogeneous minimum complex, our structures would not be 100% identical to the corresponding, diverse native complexes obtained from cells. Our structure could only suggest possible DED-DED interactions within the complexes. Additionally, our structures couldn't reveal interactions between Casp-8$^{tDED}$ trimers, the dimerization of Casp-8's protease domain, the complete structure of

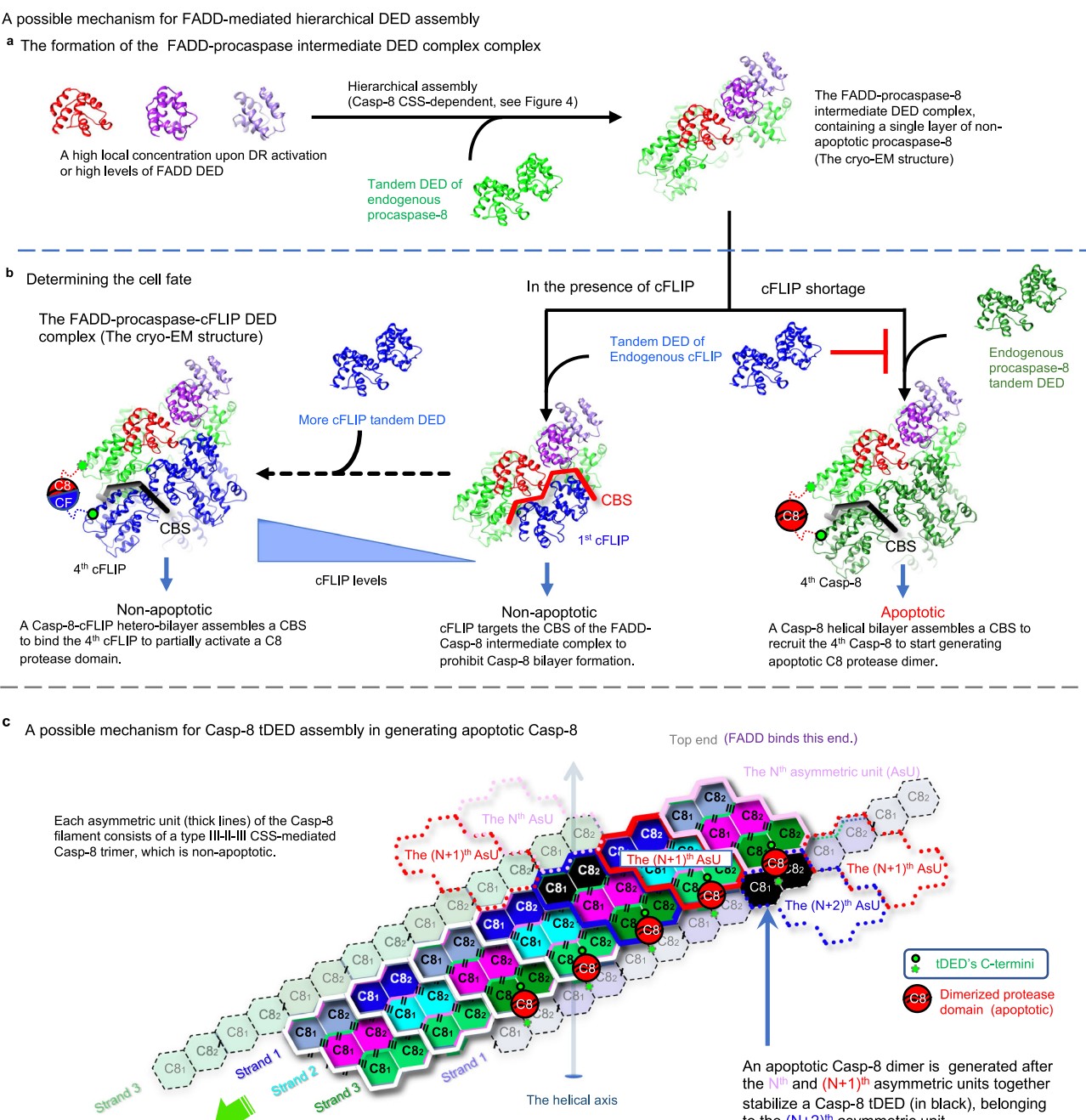

**Fig. 7 | DED assembly mechanisms in regulating apoptotic signaling.**
**a**, **b** Propose a hierarchical FADD-mediated DED assembly mechanism for the FADD-Casp-8 and FADD-Casp-8-cFLIP complexes. **a** In the first stage, the formation of a FADD-Casp-8 intermediate complex requires CSS-mediated self-assembly of endogenous Casp-8 and also a high local concentration of FADD to interact with the Casp-8 assembly. Notably, the single-FADD intermediate complex assembles in a similar manner. **b** In the second stage, cFLIP targets the CBS on the intermediate complex, terminating hierarchical apoptotic signaling. This process generates a Casp-8-cFLIP hetero-double layer, licensing local Casp-8 activation. However, when cFLIP is depleted, Casp-8 binds the same CBS, forming Casp-8 double layers and hence filaments to execute apoptosis. This unveils structurally how the levels of cFLIP mechanically regulate apoptotic signaling. **c** A 2D representation illustrates tDED assembly in a Casp-8 filament, elucidating the mechanism of generating apoptotic Casp-8. The tDED is colored as per the suggested model for EMD-11939 in Supplementary Fig. 11g. Each asymmetric unit is highlighted by thick lines. Two asymmetric units, for example, the Nth and (N + 1)th, of the Casp-8 filament assemble a Casp-8 double layer and hence a CBS, stabilizing the first Casp-8tDED (black) of the next forming (N + 2)th asymmetric unit. This arrangement allows its protease domain and the protease domain of the third Casp-8tDED located in the Nth asymmetric unit to form an apoptotic dimer (depicted as red balls). See also Supplementary Fig. 11f, g for apoptotic Casp-8 formation.

FADD because the protease domain as well as FADD DD remain invisible. However, these aspects have been addressed in prior studies[38,39,51,75–78]. Furthermore, future investigations are necessary to investigate whether cFLIP could effectively impede full Casp-8 activation if cFLIP doesn't target FADD and the first CBS on the FADD-Casp-8 intermediate complex.

In addition to our investigations into multiprotein FADD-Casp-8-cFLIP DED complexes, it is noteworthy to mention recently reported domain-swapped dimer formations in Casp-8 tDED under specific conditions, involving the F122A mutation or a two-residue truncation possibly at the C-terminus of DED2's H6[51,77]. The details of Casp-8 tDED domain-swapping merit consideration in the broader context

of DED assembly dynamics, a topic that may be explored in future studies. In a broader context of DD-fold multiprotein assembly, we found the diverse use of various CSS in self-assembly in other DD-fold multiprotein structures. For example, the RIG-I tandem CARD assembly[79] utilizes a type III-I-III CSS (comprising one type I and two type III surfaces) to self-assemble a tetramer (Supplementary Fig. 14d). In the human apoptosome[79], the heterodimeric Apaf-1-Casp-9 CARD protomer employs a type III-II-III CSS to self-assemble a trimer (Supplementary Fig. 14e). In our study, the interaction of two single FADD$^{DED}$ molecules leads to a homodimer that creates a type-III-II-III CSS surface to bind Casp-8$^{tDED}$, supporting our earlier hypothesis[80].

The structural insights provided in this study, elucidating previously undocumented interactions such as FADD$^{DED}$-FADD$^{DED}$, FADD$^{DED}$-Casp-8$^{tDED}$, FADD$^{DED}$-cFLIP$^{tDED}$, Casp-8$^{tDED}$-cFLIP$^{tDED}$, and cFLIP$^{tDED}$-cFLIP$^{tDED}$ within the ternary complex at a 3.7 Å resolution, mark a pivotal advancement in our comprehension of the intricate multiprotein DED/tDED assembly in 3D. By unveiling crucial determinants of specificity in assembling a complex with pseudo-helical symmetry, these structural data propel our insights to a higher dimension. The revelation underscores the imperative for further research to uncover the functional implications embedded in these intricate structural findings, especially in different types of cells with variations in receptor levels, signaling molecules, regulatory mechanisms, and unique genetic defects.

## Methods

### Protein expression and purification

The FADD protein fused with a C-terminal 6xHis-tag (C-tag), Casp-8$^{tDED}$ mutant containing the residues 1–185 of Casp-8, and cFLIP$^{tDED}$ protein containing the residues 1–181 of cFLIP, were individually expressed in *Escherichia coli* (BL21-CodonPlus, Agilent). Ternary complexes were co-purified by using Ni-nitrilotriacetic acid affinity resin followed by size-exclusion chromatography (SEC) using a Superdex-200 (10/300) gel-filtration column (Cytiva) equilibrated with Buffer A (80 mM NaCl and 20 mM Tris, pH 8.0). The selenomethionine (SeMet, Carbosynth FS09881)-substituted protein complex was expressed and co-purified by the same method. The cloning primers are listed in Supplementary Table 4.

### Protein crystallization

The protein complex was crystallized at a protein concentration of 10 mg/mL by the hanging-drop vapor-diffusion method at 20 °C with a solution of mixing 100 mM Na-HEPES at pH 7.0, 100 mM TBG at pH 9.0, 8% PEG8000, and 10 mM TCEP at pH 7.3. The SeMet-substituted protein complex was crystallized by the same hanging-drop vapor-diffusion method. The SeMet-substituted protein crystals was crystallized by using a solution containing 92 mM Na-HEPES at pH 7.0, 169 mM TBG at pH 9.0, 8% PEG8000, and 9 mM TCEP at pH 7.3.

### Crystal structure determination and model building

Anomalous and native diffraction data was collected at the beamline TLS BL13B1 at NSRRC, using the wavelengths of 0.97939 and 0.99984 Å, respectively, at 110 K. The data sets were processed using the program HKL-2000. The structure of the FA$^{FuL\_H9G}$-C8$^{FGLG}$-CF$^{H7G}$ complex was determined by using a SeMet-derivative dataset and the MR-SAD method in Phenix ver 1.14-3260. There was one 1:5:4 FA$^{FuL\_H9G}$-C8$^{tDED\_FGLG}$-CF$^{tDED\_H7G}$ complex per asymmetric unit with a solvent content of 65%. The structural model was iteratively built and refined using the programs Coot ver 0.8.9.1 EL and Phenix ver 1.14-3260, respectively. There is no electron density for FADD DD. All residues belong to the Ramachandran favored and allowed regions with no outliers in both structures. Data collection and refinement statistics were summarized in Supplementary Table 1.

### Structural analysis and molecular graphics

The structures were analyzed using the MUSTANG ver 3.2.3, Mol-Probity (http://kinemage.biochem.duke.edu), Coot ver 0.8.9.1 EL or WinCoot ver 0.8.2, PyMOL ver 1.8.2.1 and UCSF Chimera programs ver 1.13.1, and the qtPISA program in CCP4i2 ver 7.0.060. All interface residues and areas were identified and estimated, respectively, by qtPISA. All figures for presenting the structures were prepared by using the PyMOL ver 1.8.2.1 and UCSF Chimera program ver 1.13.1.

### The ball-and-ribbon diagram

In Figs. 2b and d, the ball-and-ribbon diagrams show the spatial relationship of individual DEDs of the ternary complexes by using the PyMOL program ver 1.8.2.1 to draw ribbons that connect the spatially conserved Cα atoms (shown as balls) of every two adjacent DEDs. The ribbons show that DEDs assemble via the type I, II, and also III interfaces: blue, green, and cyan ribbons represent the DED assembly through the type I interface (called the type I connectivity), while gray ribbons represent the DED assembly through the type II interface (the type II connectivity). Red, yellow, and pink ribbons represent the DED assembly through the type III interface (the type III connectivity). The balls are the Cα atoms of the spatially conserved leucine residues among DEDs including FADD residue L63, Casp-8 residues L62 and L162, and cFLIP residues L55 and L152 (Supplementary Fig. 1c). In addition, the spatially conserved Cα atoms were also used to calculate the helical twist shown in Fig. 5c by the HELANAL-Plus program (https://dna.mbu.iisc.ac.in/resources.htm).

### Multi-angle light scattering

In the SEC-MALS experiments, ternary DED complexes were injected onto a Superdex 200 (5/150) size-exclusion column and a Wyatt protein SEC column with the pore size of 300 Å, respectively. The SEC column was equilibrated with 20 mM Tris (pH 8.0) and 80 mM NaCl at room temperature, through a fast protein liquid chromatography system (Äkta purifier 10) coupled with a three-angle light-scattering detector (mini-DAWN TREOS) and a refractive index detector (Optilab T-rEX) (Wyatt Technology). Data analysis was carried out using the program ASTRA ver 6.0.5.3.

### Small-angle X-ray scattering

In the SEC-SAXS experiments, the SAXS data was collected at the SWAXS beamline BL23A1 at NSRRC with an online size-exclusion high-performance liquid chromatographic system (Agilent chromatographic system 1260 series). Each protein sample was injected into an Agilent Bio SEC-5 HPLC column with a flow rate of 0.35 ml/min. Once the complex is eluted, the solution will go through a quartz capillary (2.0 mm diameter) with a flow rate decreased to 0.03 ml/min for the SAXS data collected with 30 s per frame using a Pilatus 1M-F area detector. All buffer solutions were measured under the same condition for background scattering subtraction. The SAXS data was analyzed by the ATSAS program suite ver 2.7 to 3.2.1. Data collection and structural parameters were summarized in Supplementary Table 2.

### Treatment of protein precipitates for EM studies

The precipitates of overexpressed wild-type protein or complexes were resuspended in water for negative stain EM studies.

### Negative staining

In all, 4 µl sample solution was applied onto the carbon-coated side of a TEM carbon type-B grid (Ted Pella, 01814-F) for 30 seconds and then the liquid was drawn off from the edge of grid with a filter paper (Whatman). Immediately, 4 µl of 1% uranyl acetate or uranyl formate was applied to the grid for 30 seconds and drawn off with a filter paper. Then the grid was placed into an electronic dry cabinet to air dry before imaging, which uses a FEI Tecnai G2 F20 S-TWIN at ICOB,

Academia Sinica. To detect His-tagged targets, 5 nm Ni-NTA Nanogold was used according to Supplier's instruction (Nanoprobes).

## Protein sample cross-linking

We utilized the gradient-based cross-linking method, also called the GraFix method, for protein sample cross-linking. Before cross-linking, purified protein complexes were dialyzed against a HEPES buffer and then concentrated to 1 mg/ml. For creating gradients of 10–30% glycerol with 0.016-0.05% glutaraldehyde, a centrifuge tube (14 × 89 mm, Beckman Coulter) containing a top solution (80 mM NaCl, 20 mM HEPES pH 8, 10% glycerol) and a bottom solution (80 mM NaCl, 20 mM HEPES pH 8, 30% glycerol, 0.05% glutaraldehyde) was placed into a gradient mixer (the Gradient Master 108, BioComp Instrument). Subsequently, a 200-μl buffering cushion (80 mM NaCl, 20 mM HEPES pH 8, 7% glycerol) was added on the top of the gradient, followed by loading a 200-μl protein sample on the top of the cushion. The ultra-centrifugation was carried out at 4 °C in a swing-out rotor (SW-41, Beckmann) at a speed of 273,620 × g or 40,000 rpm for 18.5 h. The gradient was then fractionated from the bottom and each fraction was analyzed by SDS-PAGE (Bio-Rad). The peak fractions were further loaded onto a prewashed desalting column (Zeba™ Spin Desalting Columns, Thermo Scientific) to remove glycerol and excess cross linkers. The cross-linked complex was eluted in a TRIS buffer (80 mM NaCl, 20 mM Tris pH 8) and stored at 4 °C.

## Cryo-EM grids preparation and data acquisition

The glutaraldehyde cross-linked protein samples (~250 μg/ml) (Complex B) were applied to glow-discharged holey carbon grids (Quantifoil Cu, R1.2/1.3, 300 mesh). Each grid was blotted for 3 ~ 4 s and flash-plunged into liquid ethane pre-cooled in liquid nitrogen using an FEI Vitrobot mark IV operated at 4 °C and 100% humidity. Cryo-EM data were acquired on a Titan Krios (FEI) operated at 300 kV, equipped with a Gatan K2 Summit detector at ASCEM, Academia Sinica. The EPU software was utilized for automated data collection. Movies were collected at a nominal magnification of 165,000× in counting mode resulting in a calibrated pixel size of 0.84 Å/pixel, with a defocus range of approximately −0.5 to −2.5 μm. Eighty frames were recorded over 6 s of exposure at a dose rate of 0.9 electrons per Å$^2$ per frame. Data collection, refinement and validation statistics were summarized in Supplementary Table 3.

The non-crosslinked protein sample (Complex A) was absorbed onto freshly glow-discharged Quantifoil 2/1 grids with PELCO easiGlow glow discharger and flash frozen in liquid ethane using Leica GP Climate controlled sample plunger with a controlled temperature and humidity. Cryo-EM movies were collected with FEI Talos Arctica electron microscope equipped with a Gatan BioQuatum energy filter and a K2 Summit direct electron detector at the Institute for Quantitative Biomedicine at Rutgers University, USA using the software EPU for automated data collection. We collected 30 movie frames with a combined exposure time of 7.5 s corresponding to a total dose of 24 electrons per Å$^2$ (0.8 electrons/frame). Defocus values were set from −0.5 to −3 μm at a magnification of 130,000 times resulting with a pixel size of 1 Å.

## Cryo-EM image processing and model refinement

All datasets were processed using the CryoSparc V2 software package installed in our own computers. Movie frames were aligned by running the job Full-frame motion correction. The contrast transfer function parameters for each aligned micrograph were estimated by running CTF estimation (CTFFIND4). Particle templates were created from the particles picked manually from 15 to 20 micrographs. Particles were then picked automatically by running Template picker. All particles were local-aligned and extracted by running Local motion correction, and then subjected to 2D classification for several cycles to remove junk particles. Ab initio 3D classification was carried out by running Ab

initio Reconstruction. After removing junk particles, the jobs the heterogeneous refinement, homogeneous refinement, and non-uniform refinement were performed. All maps were sharpened and the resolution was estimated using the Fourier shell correlation (FSC) = 0.143 cutoff criterion (Supplementary Figs. 5 and 6).

The crystal structures of FADD$^{DED}$, cFLIP$^{tDED}$, and C8$^{tDED}$, were fit into the Cryo-EM envelops using UCSF Chimera ver 1.13.1 and refined in real space using the Phenix ver 1.14-3260 software. Identification of Casp-8 and cFLIP is facilitated by the presence of envelops for helix H7 of tDED, with further confirmation of cFLIP identity through the observation of disordered helix H3 in both cFLIP DED1 and DED2 (Supplementary Figs. 1c and 9). Notably, given that tDED but not FADD DED has helix H7 (Supplementary Fig. 1a and Supplementary Figs. 8, 9, vs 7), the absence of an envelope for helix H7 of tDED distinctly indicates that in Complex B, the position typically occupied by the tDED of C8d is instead taken up by two FADD DED molecules (Supplementary Fig. 2e). The Casp-8 protease domain and FADD DD were not visible.

## Mutagenesis and pulldown assays

All gene mutations were generated by using the KOD Plus Mutagenesis Kit (TOYOBO) or In-Fusion HD Cloning Plus kit (TaKaRa). The His-tagged proteins were expressed and captured by the resin (Qiagen). Excess proteins and impurities were removed by three times of wash with 40 mM imidazole. The resin was then mixed with non-tagged protein lysates at room temperature for 1 h, and then washed again with three times of 40 mM imidazole. The proteins were eluted with 1 M imidazole. All samples are subjected to SDS-PAGE analysis with Coomassie blue staining. The mutagenesis primers are listed in Supplementary Table 4.

## Cell culture

Immortalized mouse embryonic fibroblast (iMEF)[81] and HEK293T cells (American Type Culture Collection (ATCC) CRL-3216, RRID:CVCL_0063) were cultured in Dulbecco's Modified Eagle's Medium (DMEM) high glucose supplied with 10% FBS, Sodium pyruvate and penicillin/streptomycin. HAP1 cells (human chronic myeloid leukemia CML-derived HAP1 cells) were purchased from Horizon Discovery Group plc (RRID:CVCL_SM74) and cultured Iscove's Modified Dulbecco's Medium (IMDM) with 10% FBS and penicillin/streptomycin (Gibico) according to Supplier's instruction (Horizon Discovery). These cell lines were not authenticated by ourself in this study. No commonly misidentified cell lines were used in this study.

## Antibodies used in the cell-based experiments

phospho-IKKα/β (Ser176/180)(Cell Signaling, #2697 L, 1:1000); IKKα/IKKβ (Santa Cruz, #sc-7607,1:1000); GAPDH (GeneTex, #GTX627408, 1:5000); cFLIP (dave-2) (Adipogen, #AG-20B-0005, 1:1000); Casp-8 (D35G2) (Cell Signaling, #4790, 1:1000); Cleaved Casp-3 (Asp175) (5A1E) (Cell Signaling, #9664, 1:1000); PARP (Cell Signaling, 930 #9532,1:1000); phospho-MLKL (S345) (Abcam, #ab196436, 1:2000); MLKL (clone 3H1) (Millipore, #MABC604, 1:1000); phospho-RIPK3 (Thr231/Ser232) (Cell Signaling, #57220, 1:1000); RIPK3 (Cell Signaling, #95702, 1:1000); RIPK1 (BD, #610459, 1:2000); Phospho-RIPK1 (Ser166) (Cell Signaling, #31122, 1:1000); phospho-p38 (Thr180/Tyr182) (Cell Signaling, #9211,1:1000); p38 MAPK (Cell Signaling, #9212, 1:1000); phospho-JNK/SAPK (Thr183/Tyr185) (Cell Signaling, #9251, 1:1000); JNK/SAPK (Cell Signaling, #9252, 1:1000); FADD (Abcam, #Ab124812, 1:1000); FADD (human specific) (Cell Signaling, #2782, 1:1000).

## Reagents used in the cell-based experiments

Recombinant mouse TNFα (R&D Systems, #410-MT), Recombinant human TNFα (R&D Systems, #210-TA), SuperKillerTRAIL (Adipogen, #AG-40T-0004-C020), SM-406 (ApexBio, #A3019), z-VAD-fmk

(ApexBio, #A1902), Nec-1s (Biovision, #2263), Doxycycline hyclate (Sigma, #D9891), Cyclohexylamine (Sigma, #C1988).

## Generation of FADD knockout cells

SgRNA targeting on *FADD* and pAll-Cas9.pPuro plasmid were designed and purchased from the National RNAi Core Facility at Academia Sinica, Taiwan. iMEF was transfected with CRISPR plasmid using Turbofect (Thermo) according to manufacturer's instructions. On the third day after transfection, iMEF underwent serial dilutions and antibiotic selection to remove unwanted cells. Surviving cell colonies were examined by western blotting to confirm the gene knockout (Supplementary Fig. 15a). The FADD-gene knockout colonies were expanded and stored for the following experiments. *FADD*-KO clone sg4.B9 was used for expansion because necroptotic signaling molecules MLKL and RIPK3 remain there (Supplementary Fig. 15a). TNF-induced NF-κB activation signaling remains normal in the *FADD*-KO iMEF cells (clone sg4.B9) with or without CHX (Supplementary Fig. 15g, h). The FADD-gene knockout HAP1 (RRID:CVCL_SM74) in Fig. 6g were purchased from Horizon Discovery.

## FADD reconstitution

The human or mouse full-length FADD gene was cloned into the pAS4.1w.Pbsd-aOn vector. HEK293T cells (American Type Culture Collection (ATCC) CRL-3216, RRID:CVCL_0063) were transfected with *FADD* constructs and the packaging plasmids pMD.G and pCMVR8.91 (the National RNAi Core Facility at Academia Sinica) with Turbofect according to manufacturer's instructions. Virus-containing mediums were harvested 48 and 72 h after transfection. FADD-gene knockout iMEF clone was incubated with the virus-containing medium overnight for infection. After infection, iMEF were selected with blasticidin (5 μg/ml) to eliminate uninfected cells. For HAP1 cells, 7.5 μg/ml blasticidin was used. Stable colonies were then saved for following assays. FADD protein expression was induced by doxycycline hyclate (0.1–1 μg/ml) for 16–24 h before additional treatments. The cloning primers are listed in Supplementary Table 4.

## Cell-based mutagenesis assays

Tet-inducible plasmids containing FADD WT or mutant sequences were transfected into FADD-deficient immortalized mouse embryonic fibroblast (FADD⁻/⁻ iMEF) using a lentivirus packaging system. Mouse FADD (mFADD)-reconstituted iMEF were treated with doxycycline overnight to induce FADD expression. iMEF were seeded and grew to reach 60-80% confluence before assays. To induce apoptosis, iMEF were pretreated with cycloheximide (CHX, 10 μg/ml) for 30 min, followed by the stimulation with recombinant mouse TNFα (10 ng/ml) for 6 h or the indicated time. For necroptosis induction, iMEF were pretreated with cycloheximide (10 μg/ml) and z-VAD-fmk (50 μM), before TNFα treatment. Human FADD (hFADD)-reconstituted HAP1 cells were treated with doxycycline overnight to induce FADD expression. To induce apoptosis in HAP1 cell, HAP1 cells were pretreated with cycloheximide (10 μg/ml) for 30 min, followed by the stimulation with SuperKillerTRAIL (50 ng/ml). All the mutants were created by KOD-based site-direct mutagenesis by using the primers listed in Supplementary Table 4. Uncropped data are included in a Source Data file.

## Fluorescence apoptosis and Necrosis assay

iMEFs were seeded in a 96-well plate (5000 cell per well) a day before the assay. To induce apoptosis, iMEF were pretreated with cycloheximide (CHX, 10 μg/ml) for 30 min, followed by the stimulation with recombinant mouse TNFα (10 ng/ml) for 6 h or the indicated time. For necroptosis induction, iMEF were pretreated with cycloheximide (10 μg/ml) and z-VAD-fmk (50 μM), before TNFα treatment. Nucleic acid and the Casp-3 activity were detected by using SYTO green (5 μM) and DEVD-AMC (20 μM), respectively. Fluorescent signal was detected

by a 96-well plate reader (Victor X, PerkinElmer) at the time indicated in the figures and the data was analyzed. Bar chars were generated by GraphPad Prism 9.4.1. Notably, this assay was modified from a previous assay[82].

## Cell lysate-based mutagenesis study

In vitro mutagenesis experiments to test FADD-mediated signaling were performed by a HeLa cell lysate-based protein expression system (1-Step Human Coupled IVT kit, Thermo). The full-length wild type and mutant FADD proteins were expressed by PURExpress in vitro protein synthesis kit (NEB) using pET-26 vector. The reaction mixture with expressed FADD protein was added to the reaction component of the IVT kit to a final FADD concentration of about 4 μM (measured by UV absorbance), higher than -0.33 μM of endogenous FADD[83]. The mixture was incubated at 30 °C for 6 h and then was analyzed by SDS-PAGE and western blotting. All the mutants were created by KOD-based site-direct mutagenesis by using the primers listed in Supplementary Table 4. Uncropped data are included in a Source Data file.

## Antibodies used in the cell lysate-based experiments

FLIP (D5J1E) (Cell Signaling, #56343, 1:2000); Casp-8(D35G2) (Cell Signaling, #4790, 1:4000); Casp-3 antibody (Cell Signaling, #9662, 1:4000); Cleaved Casp-3 (Asp175) (5A1E) (Cell Signaling, #9664, 1:2000); Human FADD (Cell Signaling, #2782, 1:2000); PARP (46D11) (Cell Signaling, #9532,1:2000); RIPK1 (Cell Signaling, #4926, 1:2000); Phospho-RIPK1 (Ser166) (D1L3S) (Cell Signaling, #65746, 1:2000); GAPDH (14C10) (HRP Conjugate) (Cell Signaling, #3683, 1:4000); Goat Anti-Rabbit IgG H&L (HRP) (Abcam, #ab6721, 1:10,000).

## Reporting summary

Further information on research design is available in the Nature Portfolio Reporting Summary linked to this article.

## Data availability

The atomic coordinates and reflection files for the crystal structures generated in this study have been deposited in the Worldwide Protein Data Bank (wwPDB) under accession codes 8YD7 (The SeMet derivative of the single-FADD-Casp-8-cFLIP DED complex) and 8YD8 (Native single-FADD-Casp-8-cFLIP DED complex). The cryo-EM structures generated in this study have been deposited in the Electron Microscopy Data Bank (EMDB) under accession codes EMD-39126 (The triple-FADD-Casp-8-cFLIP DED complex B) and EMD-39127 (The triple-FADD-Casp-8-cFLIP DED complex A). The atomic coordinates for the cryo-EM structure generated in this study have been deposited in the wwPDB under accession code 8YBX (The triple-FADD-Casp-8-cFLIP DED complex B). The cryo-EM structure used in this study are available in the EMDB under accession codes EMD-11939 (Central region of the Caspase-8-FADD complex), and EMD-11941 (The ternary complex of full-length Caspase-8 with FADD and FLIPs). The atomic coordinates for the cryo-EM structure used in this study are available in the wwPDB under accession codes 5L08[38] (Casp-8 tDED filaments). All other data generated in this study are provided in the Supplementary Information or a Source Data file. Alternatively, any data that support this study are also available from the corresponding authors upon request. Source data are provided with this paper.

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

## Acknowledgements

This work was supported by Academia Sinica (AS) Thematic Program (AS-TP-107-L16, AS-TP-107-L16-1, AS-102-TP-B14 and AS-102-TP-B14-2 to S.-C.L.; AS-TP-107-L16-2 and AS-102-TP-B14-1 to Y.-C.L; AS-TP-107-L16-3 to L.-C.H.; AS Postdoc Fellowships to C.-Y.Y. and T.-W.S.), Ministry of Science and Technology (MoST) (MoST 107-2320-B-001-018-, 108-2311-B-001-018-, 109-2311-B-001-016-, and 110-2311-B-001-015- to S.-C.L.; MoST 107-2320-B-006-062-MY3, and 111-2311-B-006-005-MY3 to Y.-C.L.; MoST 108-2320-B-002-020-MY3, 111-2320-B-002-048-MY3, and 112-2326-B-002-007- to L.-C.H.), and Taiwan Protein Project (TPP) (Grant No. AS-KPQ-105-TPP to Y.-C.L and S.-C.L.). We thank the Protein X-ray Diffractometer, Chameleon, and Vitrobot facilities of GRC for crystallization screening and sample preparation for cryo-EM, DNA Sequencing Core Facility of IBMS (AS CFII Project AS-CFII-113-A12) for DNA sequencing, AS Biological EM Core Facility of ICOB (AS-CFII-111-203) for negative stain EM data, AS Cryo-EM Facility (ASCEM) (AS-CFII-111-210 and TPP AS-KPQ-109-TPP2) for cryo-EM data. We thank the beamline BL13B1 and BL23A1 at NSRRC, Taiwan, for X-ray diffraction and SAXS data, respectively. We thank Prof. Ming-Daw Tsai and Prof. Chih-Hao Lee for their insightful feedback.

## Author contributions

C.-Y.Y. expressed, purified and crystallized the complexes, collected the SAXS, X-ray, and cryo-EM data and solved the structures. C.-I.L. did the cell-based experiments. Y.-C.T. did the cell lysate-based experiments. Y.-C.T., Y.-F.T. and Y.-C.Lu did the pulldown experiments. A.W.K. collected the cryo-EM data of the complex A and helped to solve the

structure. C.-Y.Y., T.-W.S. and Y.-T.W. help analyzing the data. C.-Y.Y., C.-I.L., Y.-C.T., Y.-F.T., Y.-C.Lu and Y.-T.W. made the constructs. L.-C.H., Y.-C.Lo and S.-C.L. supervised the project. S.-C.L. and Y.-C.Lo analyzed and interpreted the data, made the final figures, and wrote the manuscript. A.W.K., Y.-C.Lo and L.-C.H. provided comments and revised the manuscript. S.-C.L. initiated the project.

## Competing interests

The authors declare no competing interests.
