## [Peer Review File · Nature Communications]

Deciphering DED Assembly Mechanisms in FADD-
Procaspase-8-cFLIP Complexes Regulating ApoptosisEditorial Note: This manuscript has been previously reviewed at another journal that is not operating a transparent peer review scheme. This document only contains reviewer comments and rebuttal letters for versions considered at *Nature Communications*.

REVIEWERS' COMMENTS

Reviewer #1 (Remarks to the Author):

The authors have carefully considered my concerns and addressed them in a well crafted response.

Reviewer #2 (Remarks to the Author):

The authors have addressed all my comments satisfactorily.

Reviewer #3 (Remarks to the Author):

I support the publication of this paper because the authors responded appropriately to all my comments and concerns.

Response to Reviewers' comments:

Reviewer #1 (Remarks to the Author):

The authors have carefully considered my concerns and addressed them in a well-crafted response.

Response: We thank the valuable feedback from Reviewer #1. We had carefully considered the concerns raised by Reviewer #1 and had addressed them comprehensively in the revised manuscript.

Reviewer #2 (Remarks to the Author):

The authors have addressed all my comments satisfactorily.

Response: We are pleased to hear that Reviewer #2 found our responses satisfactory. We have incorporated all suggested changes to enhance the clarity and coherence of the manuscript.

Reviewer #3 (Remarks to the Author):

I support the publication of this paper because the authors responded appropriately to all my comments and concerns. "

Response: We are grateful for Reviewer #3's support and positive feedback. We have ensured that all comments and concerns raised by Reviewer #3 have been adequately addressed in the revised manuscript